# Bayesian Kernel Regression for Functional Data

## Abstract

In supervised learning, the output variable to be predicted is often represented as a function, such as a spectrum or probability distribution. Despite its importance, functional output regression remains relatively unexplored. In this study, we propose a novel functional output regression model based on kernel methods. Unlike conventional approaches that independently train regressors with scalar outputs for each measurement point of the output function, our method leverages the covariance structure within the function values, akin to multitask learning, leading to enhanced learning efficiency and improved prediction accuracy. Compared with existing nonlinear function-on-scalar models in statistical functional data analysis, our model effectively handles high-dimensional nonlinearity while maintaining a simple model structure. Furthermore, the fully kernel-based formulation allows the model to be expressed within the framework of reproducing kernel Hilbert space, providing an analytic form for parameter estimation and a solid foundation for further theoretical analysis. The proposed model delivers a functional output predictive distribution derived analytically from a Bayesian perspective, enabling the quantification of uncertainty in the predicted function. We demonstrate the model's enhanced prediction performance through experiments on artificial datasets and density of states prediction tasks in materials science.

## 1 Introduction

In conventional supervised learning, predictive models typically map vectorized inputs to scalar or vector outputs Nasteski (2017); Crisci et al. (2012). However, in many real-world applications, output variables are expressed as functions. This is particularly common in materials research, where outputs often take the form of functional values, such as spectra or probability distributions Iwayama et al. (2022). For instance, a physical property to be predicted often depends on measurement conditions, such as temperature and pressure, with the output represented as a function over these conditions. Conditional image generation tasks can also be reduced to regression problems with output variables represented as bivariate or trivariate functions, for example, material microscopic images. Motivated by potential applications in materials science, we develop a kernel-based functional output regression model. Our method addresses problem settings where the output variable $Y(X, t)$ is a function over $t \in \mathbb{R}^q$, dependent on a covariate $X$. Unlike conventional approaches that independently train scalar output regressors for each measurement point of the output function, our method exploits the covariance structure and smoothness prior inherent in the function values, similar to multitask learning, leading to enhanced learning efficiency.

Functional data analysis (FDA) is a statistical framework where the output variable $Y$ or its covariate $X$ is given as a function Ramsay & Silverman (2005). Within FDA, extensive research has focused on functionalizing discrete data as functional data (i.e., converting discrete data into explicit continuous functions) and performing regression under the assumption that outputs or covariates have undergone this functionalization process beforehand Ullah & Finch (2013); Wang et al. (2016); Li et al. (2022). In this study, we aimed to obtain a functional output model directly from vector input values and discretely observed functional data, bypassing the functionalization process. In FDA, methods addressing such tasks are referred to as function-on-scalar regression (FSR).

Within FSR, a functional linear model (FLM), which extends a linear model to a functional output, is the most extensively studied Chen et al. (2016); Luo & Qi (2023). In FLM, the functional output $Y(X, t)$ for an input value $X = (x_1, \cdots, x_p)^\top$ is modeled as $Y(X, t) = \beta_0(t) + \sum_{j=1}^{p} x_j \beta_j(t) + \epsilon$, where $\beta_j(t)$ represents basis functions, and $x_j$s are the covariates. As evident from this formulation, FLM is linear with respect to the input value $X$, imposing significant limitations on the model's representational capability. However, despite the need for greater flexibility, research on nonlinear FSR has been relatively limited. Existing FSR methods include the functional quadratic regression, which employs an additional second-order term for covariate $x_j$ Yao & Müller (2010); the functional additive mixture model, which simply extends an additive mixture model to functional outputs Scheipl et al. (2015); and a neural network-based model that leverages the functional universal approximation theorem Luo & Qi (2023).

In this study, we propose kernel regression for functional data (KRFD) that naturally incorporates nonlinearity by introducing a kernel function for the covariate $X$. Leveraging the kernel trick, KRFD can express nonlinearity with respect to covariates $X$, while maintaining a simple model structure. This approach was motivated by a previous study Iwayama et al. (2022). In that work, the functional output $Y(X, t)$ is modeled as $Y(X, t) = \sum_{l=1}^{L} k_T(t, t_l) c_l(X) + \mu(t) + \epsilon$, where kernel functions $k_T(t, t_l)$ are placed on grid points $\{t_l \mid l = 1, \ldots, L\}$ in the $t$-space, $c_l(X)$s represent their coefficients dependent on $X$, and $\mu(t)$ is the baseline function. The coefficients $c_l(X)$ were modeled using a neural network (NN). By contrast, KRFD simplifies this model by replacing the NN with a kernel ridge regression (KRR) model, expressed as $c_l(X) = \sum_{n=1}^{N} \theta_{nl} k_G(X, X_n)$. This substitution makes the model linear with respect to the unknown parameters $\theta_{nl}$, enabling the derivation of analytical expressions for parameter estimation, exact Bayesian inference without approximations, and a mathematical formulation within the framework of reproducing kernel Hilbert spaces (RKHS). These capabilities lay a solid foundation for further theoretical analysis. Specifically, the Bayesian extension allows for the analytical evaluation of prediction uncertainty. Moreover, linking the model to the RKHS framework reveals a connection between the KRFD model and existing multitask learning (MTL) based on the assumption of separable kernels Bonilla et al. (2007); Ciliberto et al. (2015).

## 2 Methods

### 2.1 Models

Consider the regression problem of predicting the function $Y(X, t) \in \mathbb{R}$ over $t \in \mathbb{R}^q$ from the vector input values $X \in \mathbb{R}^p$. The KRFD is expressed as follows:

$$Y(X, t) = f(X, t) + \mu(X) + \epsilon, \tag{1}$$

where $f(X, t) \in \mathbb{R}$ denotes an arbitrary function on $X$ and $t$, $\mu(X)$ denotes an baseline function depedent only on $X$, and $\epsilon$ is the measurement noise.

We assume that the output values $Y(X_i, t_j) \in \mathbb{R}$ are measured for all combinations of $X_i \in \mathbb{R}^p (i = 1, \ldots, N)$ and $t_j \in \mathbb{R}^q (j = 1, \ldots, L)$. This implies that all function values $Y(X_i, t)(i = 1, \ldots, N)$ are observed at the same measurement points $t_j$ $(j = 1, \ldots, L)$. In later sections, we consider the case where the measurement points differ across different $X_i$s.

In KRFD, $f(X, t)$ is modeled by a linear combination of positive definite kernels Fasshauer (2011) defined in the $t$-space, $k_T(t, t')$, as follows:

$$f(X, t) = \sum_{l=1}^{L} c_l(X) k_T(t, t_l), \tag{2}$$

where $c_l(X)$ is an $X$-dependent coefficient function. According to Eq.2, the $L$ kernel functions $k_T(t, \cdot)$ are centered at $t_1, \ldots, t_L$, and the weight of the kernel function is controlled by the function $c_l(X)(l = 1, \ldots, L)$.

In KRFD, the function $c_l(X)$ is represented by a linear combination of positive definite kernels defined in the $X$-space, $k_G(X, X')$, as follows:

$$c_l(X) = \sum_{n=1}^{N} \theta_{nl} k_G(X, X_n) \;\; (l = 1, \ldots, L), \tag{3}$$

where $\theta_{nl}$ $(n = 1, \ldots, N, l = 1, \ldots, L)$ are the trainable parameters. By substituting Eq.3 in Eq.2, the function $f(X, t)$ is expressed as follows:

$$f(X, t) = \sum_{n=1}^{N} \sum_{l=1}^{L} k_G(X, X_n) \theta_{nl} k_T(t, t_l) \tag{4}$$

Similarly, the $t$-independent term $\mu(X)$ is represented by a linear combination of positive definite kernels defined in the $X$-space $k_M(X, X)$ as follows:

$$\mu(X) = \sum_{m=1}^{N} c_m k_M(X, X_m). \tag{5}$$

where $c_m$ $(m = 1, \ldots, N)$ are the trainable parameters; By substituting Eqs.4 and 5 in Eq.1, we obtain the basic form of the model as follows:

$$Y(X, t) = \sum_{n=1}^{N} \sum_{l=1}^{L} k_G(X, X_n) \theta_{nl} k_T(t, t_l) \; + \; \sum_{m=1}^{N} c_m k_M(X, X_m) + \epsilon. \tag{6}$$

As shown in Eq. 6, the KRFD model is linear with respect to the trainable parameters $\theta_{nl}, c_m$. Nonlinearities are incorporated into the model only through the kernel functions. Figure 1 shows the schematic of the model structure.

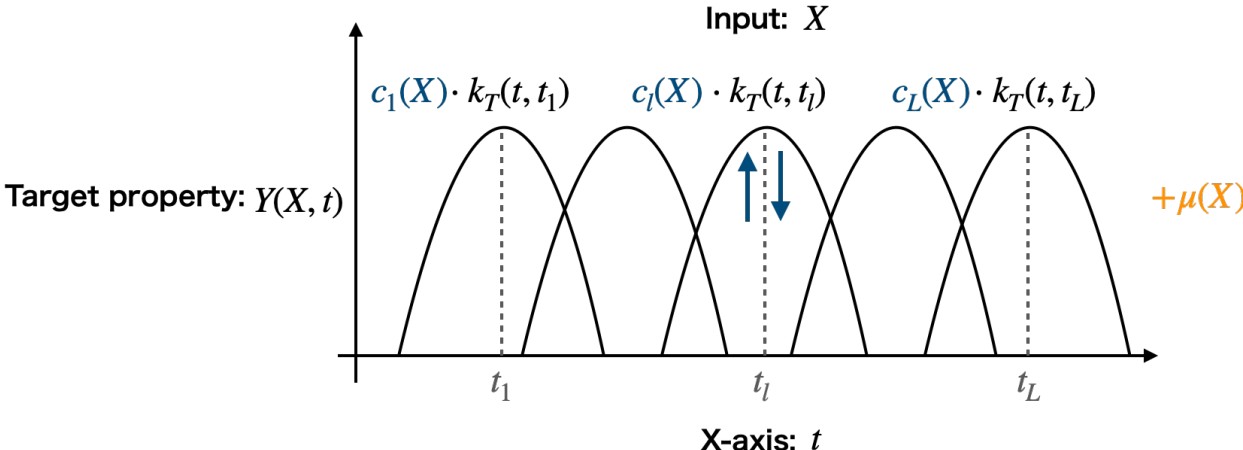

Figure 1: Schematic of the KRFD models, showing how the function output $Y(X, t)$ is modeled. The bell-shaped functions represent the kernel functions $k_T(\cdot, \cdot)$, centered at the measurement points $t_1, \ldots, t_L$. Their weights $c_l(X)$ are $X$-dependent functions modeled by kernel regressors. The intercept term $\mu(X)$ is included to capture input-dependent shifts independent of $t$.

## 2.2 Bayesian estimation

In this study, the KRFD models are estimated within the framework of Bayesian inference. Here, it is assumed that the measurement noise $\epsilon$ is distributed independently and identically according to a normal distribution

$N(0, \sigma^2)$ with mean 0 and variance $\sigma^2$. Then, the training data $Y(X_i, t_j)(i = 1, \ldots, N, j = 1, \ldots, L)$ is generated as follows:

$$Y(X_i, t_j) = \sum_{n=1}^{N} \sum_{l=1}^{L} k_G(X_i, X_n)\theta_{nl}k_T(t_j, t_l) + \sum_{m=1}^{N} c_m k_M(X_i, X_m) + \epsilon_{ij},$$

$$\epsilon_{ij} \overset{i.i.d}{\sim} N(0, \sigma^2), \quad (i = 1, \ldots, N, \ j = 1, \ldots, L). \tag{7}$$

Let $\boldsymbol{G}, \boldsymbol{T}$, and $\boldsymbol{M}$ denote gram matrices with entries defined as $\boldsymbol{G}_{ij} = k_G(X_i, X_j)$, $\boldsymbol{T}_{ij} = k_T(t_i, t_j)$, and $\boldsymbol{M}_{ij} = k_M(X_i, X_j)$, respectively. The model parameters and observed output values are expressed in vector form as $\boldsymbol{\theta} = (\theta_{11}, \cdots, \theta_{1L}, \theta_{21}, \cdots, \theta_{2L}, \cdots, \theta_{N1}, \cdots, \theta_{NL})^\top$, $\boldsymbol{c} = (c_1, \cdots, c_N)^\top$, and $\boldsymbol{y} = (Y(X_1, t_1), \cdots, Y(X_1, t_L), Y(X_2, t_1), \cdots, Y(X_2, t_L), \cdots, Y(X_N, t_1), \cdots, Y(X_N, t_L))^\top$, respectively. Using these notations, Eq. 7 can be rewritten as

$$\boldsymbol{y} = \boldsymbol{G} \otimes \boldsymbol{T} \cdot \boldsymbol{\theta} + \boldsymbol{M} \otimes \boldsymbol{1} \cdot \boldsymbol{c} + \boldsymbol{\epsilon}, \tag{8}$$

where $\otimes$ is the Kronecker product, $\boldsymbol{1}$ denotes a $L$-dimensional vector whose entries are all one, and $\boldsymbol{\epsilon}$ denotes an $NL$-dimensional random vector whose entries are distributed according to $N(0, \sigma^2)$. In summary, the likelihood of KRFD, $p(\boldsymbol{y}|\boldsymbol{\theta}, \boldsymbol{c}, \sigma^2)$, is expressed as

$$p(\boldsymbol{y}|\boldsymbol{\theta}, \boldsymbol{c}, \sigma^2) = N(\boldsymbol{G} \otimes \boldsymbol{T} \cdot \boldsymbol{\theta} + \boldsymbol{M} \otimes \boldsymbol{1} \cdot \boldsymbol{c}, \ \sigma^2 \boldsymbol{I}_{NL}), \tag{9}$$

where $\boldsymbol{I}_{NL}$ is an $NL \times NL$ identity matrix.

The prior distributions of model parameters $\boldsymbol{\theta}$, and $\boldsymbol{c}$ are defined as follows:

$$p(\boldsymbol{\theta}) = N\big(\boldsymbol{0}, \ \big(2\lambda_a(\boldsymbol{G} \otimes \boldsymbol{T}^2) + 2\lambda_b(\boldsymbol{G}^2 \otimes \boldsymbol{T}) + 2\lambda_c(\boldsymbol{G} \otimes \boldsymbol{T})\big)^{-1}\big), \tag{10}$$

$$p(\boldsymbol{c}) = N\big(\boldsymbol{0}, \ \big(2L\lambda_d \boldsymbol{M}\big)^{-1}\big), \tag{11}$$

where $\boldsymbol{0}$ is a vector whose entries are all zero and $\lambda_a, \lambda_b, \lambda_c$, and $\lambda_d$ are hyperparameters that control the dispersion of the normal distributions. As will be described later, the prior $p(\boldsymbol{\theta})$ is designed to derive the posterior $p(\boldsymbol{\theta}|\boldsymbol{y}, \boldsymbol{c}, \sigma^2)$ with a computational complexity of $\max\big(\mathcal{O}(N^3), \mathcal{O}(L^3)\big)$ without using any special transformations.

Finally, a prior distribution is prepared for the variance in the measurement noise $\sigma^2$. Here, we use the inverse gamma distribution, which is the conjugate prior to the normal distribution with a known mean.

$$p(\sigma^2) = IG(\alpha, \beta), \tag{12}$$

where $IG(\alpha, \beta)$ denotes an inverse gamma distribution with shape $\alpha$ and scale $\beta$ parameters.

In this case, the full conditional posterior probability distributions for parameters $\boldsymbol{\theta}$, $\boldsymbol{c}$, and $\sigma^2$ are expressed as follows:

$$p(\boldsymbol{\theta}|\boldsymbol{y}, \boldsymbol{c}, \sigma^2) = N(\boldsymbol{\mu}_\theta, \ \boldsymbol{\Sigma}_\theta),$$
$$\boldsymbol{\mu}_\theta = (\boldsymbol{G} + \lambda_G \boldsymbol{I}_N)^{-1} \otimes (\boldsymbol{T} + \lambda_T \boldsymbol{I}_L)^{-1} \cdot (\boldsymbol{y} - \boldsymbol{M} \otimes \boldsymbol{1} \cdot \boldsymbol{c}),$$
$$\boldsymbol{\Sigma}_\theta = \sigma^2 \cdot (\boldsymbol{G}^2 + \lambda_G \boldsymbol{G})^{-1} \otimes (\boldsymbol{T}^2 + \lambda_T \boldsymbol{T})^{-1}, \tag{13}$$
$$p(\boldsymbol{c}|\boldsymbol{y}, \boldsymbol{\theta}, \sigma^2) = N(\boldsymbol{\mu}_c, \ \boldsymbol{\Sigma}_c),$$
$$\boldsymbol{\mu}_c = 1/L \cdot (\boldsymbol{M}^2 + \lambda_M \boldsymbol{M})^{-1} \cdot \boldsymbol{M} \otimes \boldsymbol{1}^\top \cdot (\boldsymbol{y} - \boldsymbol{G} \otimes \boldsymbol{T} \cdot \boldsymbol{\theta}),$$
$$\boldsymbol{\Sigma}_c = \sigma^2/L \cdot (\boldsymbol{M}^2 + \lambda_M \boldsymbol{M})^{-1}, \tag{14}$$
$$p(\sigma^2|\boldsymbol{y}, \boldsymbol{\theta}, \boldsymbol{c}) = IG(\alpha + NL/2, \ \beta + 1/2\|\boldsymbol{y} - \boldsymbol{G} \otimes \boldsymbol{T} \cdot \boldsymbol{\theta} - \boldsymbol{M} \otimes \boldsymbol{1} \cdot \boldsymbol{c}\|_F^2), \tag{15}$$

where $\boldsymbol{I}_N$ and $\boldsymbol{I}_L$ are $N \times N$ and $L \times L$ identity matrices, respectively. Furthermore, the hyperparameters $\lambda_a, \lambda_b, \lambda_c$, and $\lambda_d$ in Eqs. 10 and 11 are substituted as $\lambda_a = \lambda_G/2\sigma^2$, $\lambda_b = \lambda_T/2\sigma^2$, $\lambda_c = \lambda_G\lambda_T/2\sigma^2$, and $\lambda_d = \lambda_M/2\sigma^2$, resulting in Eq. 13-15.

In Eqs 13 and 14, the posterior distributions $p(\boldsymbol{\theta}|\boldsymbol{y}, \boldsymbol{c}, \sigma^2)$ and $p(\boldsymbol{c}|\boldsymbol{y}, \boldsymbol{\theta}, \sigma^2)$ are conditioned by $\boldsymbol{c}$ and $\boldsymbol{\theta}$, respectively. As a preliminary step toward deriving an analytic solution for the Bayes estimator, we first consider finding the maximum posteriori (MAP) estimators for each conditional poterior as follows:

$$\boldsymbol{\theta} = \underset{\boldsymbol{\theta}}{\operatorname{argmax}}\, p(\boldsymbol{\theta}|\boldsymbol{y}, \boldsymbol{c}, \sigma^2) = \boldsymbol{\mu}_\theta = (\boldsymbol{G} + \lambda_G \boldsymbol{I}_N)^{-1} \otimes (\boldsymbol{T} + \lambda_T \boldsymbol{I}_L)^{-1} \cdot (\boldsymbol{y} - \boldsymbol{M} \otimes \boldsymbol{1} \cdot \boldsymbol{c}), \tag{16}$$

$$\boldsymbol{c} = \underset{\boldsymbol{c}}{\operatorname{argmax}}\, p(\boldsymbol{c}|\boldsymbol{y}, \boldsymbol{\theta}, \sigma^2) = \boldsymbol{\mu}_c = 1/L \cdot (\boldsymbol{M}^2 + \lambda_M \boldsymbol{M})^{-1} \cdot \boldsymbol{M} \otimes \boldsymbol{1}^\top \cdot (\boldsymbol{y} - \boldsymbol{G} \otimes \boldsymbol{T} \cdot \boldsymbol{\theta}). \tag{17}$$

Then, the unconditional MAP estimator is obtained by solving Eqs. 16 and 17, expressed as

$$\boldsymbol{\theta}_{MAP} = (\boldsymbol{G} + \lambda_G \boldsymbol{I}_N)^{-1} \otimes (\boldsymbol{T} + \lambda_T \boldsymbol{I}_L)^{-1} \cdot (\boldsymbol{y} - \boldsymbol{M} \otimes \boldsymbol{1} \cdot \boldsymbol{c}_{MAP}), \tag{18}$$

$$\boldsymbol{c}_{MAP} = (\boldsymbol{I}_N - \boldsymbol{1}^\top \boldsymbol{B} \boldsymbol{1} \boldsymbol{C} \boldsymbol{A} \boldsymbol{M})^{-1} \boldsymbol{C} (\boldsymbol{Y} - \boldsymbol{A} \boldsymbol{Y} \boldsymbol{B}) \boldsymbol{1}, \tag{19}$$

where $\boldsymbol{A} = \boldsymbol{G}(\boldsymbol{G} + \lambda_G \boldsymbol{I}_N)^{-1}$ and $\boldsymbol{B} = (\boldsymbol{T} + \lambda_T \boldsymbol{I}_L)^{-1}\boldsymbol{T}, \boldsymbol{C} = (\boldsymbol{M} + \lambda_M \boldsymbol{I}_N)^{-1}\frac{1}{L}$.

The estimators $\boldsymbol{\theta}_{MAP}$ and $\boldsymbol{c}_{MAP}$ in Eqs. 18 and 19 are no longer dependent on $\boldsymbol{c}$ and $\boldsymbol{\theta}$; the posterior distributions $p(\boldsymbol{\theta}|\boldsymbol{y}, \boldsymbol{c}, \sigma^2) = N(\boldsymbol{\mu}_\theta, \boldsymbol{\Sigma}_\theta)$ and $p(\boldsymbol{c}|\boldsymbol{y}, \boldsymbol{\theta}, \sigma^2) = N(\boldsymbol{\mu}_c, \boldsymbol{\Sigma}_c)$ can be rewritten as

$$p(\boldsymbol{\theta}|\boldsymbol{y}, \sigma^2) = N(\boldsymbol{\theta}_{MAP},\ \boldsymbol{\Sigma}_\theta), \tag{20}$$

$$p(\boldsymbol{c}|\boldsymbol{y}, \sigma^2) = N(\boldsymbol{c}_{MAP},\ \boldsymbol{\Sigma}_c). \tag{21}$$

where $\boldsymbol{\theta}_{MAP} = \boldsymbol{\mu}_\theta$ and $\boldsymbol{c}_{MAP} = \boldsymbol{\mu}_c$.

Accordingly, the variance $\sigma^2$ of the measurement error is estimated as the MAP solution for the posterior distribution of $\sigma^2$ as

$$\sigma^2_{MAP} = \underset{\sigma^2}{\operatorname{argmax}}\, p(\sigma^2|\boldsymbol{y}, \boldsymbol{\theta}_{MAP}, \boldsymbol{c}_{MAP}) = \frac{2\beta + \|\boldsymbol{y} - \boldsymbol{G} \otimes \boldsymbol{T} \cdot \boldsymbol{\theta}_{MAP} - \boldsymbol{M} \otimes \boldsymbol{1} \cdot \boldsymbol{c}_{MAP}\|_2^2}{2\alpha + 2 + NL}, \tag{22}$$

where $\|\cdot\|_2$ denotes the $\ell_2$ norm.

Furthermore, the prediction distribution for a new input $(X_{new}, t_{new})$ is analytically calculated as

$$p\big(\hat{Y}(X_{new}, t_{new})|\boldsymbol{y}, \sigma^2_{MAP}\big) = N(\mu_{pred},\ \sigma^2_{pred}),$$

$$\mu_{pred} = (\boldsymbol{g}_{new} \otimes \boldsymbol{t}_{new})^\top \boldsymbol{\theta}_{MAP} + \boldsymbol{m}_{new}^\top \boldsymbol{c}_{MAP}$$

$$\sigma^2_{pred} = \sigma^2_{MAP} \cdot \boldsymbol{g}_{new}^\top (\boldsymbol{G}^2 + \lambda_G \boldsymbol{G})^{-1} \boldsymbol{g}_{new} \cdot \boldsymbol{t}_{new}^\top (\boldsymbol{T}^2 + \lambda_T \boldsymbol{T})^{-1} \boldsymbol{t}_{new}$$

$$+\, \sigma^2_{MAP}/L \cdot \boldsymbol{m}_{new}^\top (\boldsymbol{M}^2 + \lambda_M \boldsymbol{M})^{-1} \boldsymbol{m}_{new}. \tag{23}$$

where $\boldsymbol{g}_{new} = (k_G(X_{new}, X_1), \dots, k_G(X_{new}, X_N))^\top$, $\boldsymbol{t}_{new} = (k_T(t_{new}, t_1), \dots, k_T(t_{new}, t_L))^\top$, and $\boldsymbol{m}_{new} = (k_M(X_{new}, X_1), \dots, k_M(X_{new}, X_N))^\top$.

## 2.3 Extension to sparse functional data

This section presents an extension of the KRFD tailored for sparse functional data, referred to as the KRSFD (kernel regression for sparse functional data). We assume that the functional output values $Y(X_i, t) \in \mathbb{R}$ are observed at different measurement points $t_{ij} \in \mathbb{R}^q (j = 1, \dots, N_i)$ across different inputs $X_i \in \mathbb{R}^p (i = 1, \dots, N)$. The KRSFD is then defined as follows:

$$Y(X, t) = \sum_{n=1}^{N} \sum_{l=1}^{L} k_G(X, X_n)\theta_{nl} k_T(t, t_l) + \epsilon, \tag{24}$$

where $\{t_1, \dots, t_L\}$ represents the kernel centers of $k_T(t, \cdot)$. In KRFD, the kernel centers are the fixed measurement point set $t_j \in \mathbb{R}^q (j = 1, \dots, L)$ provided by the training data. However, it is assumed here that they can be specified at any points. For simplicity, in Eq. 24, the $t$-independent term $\mu(X)$ is omitted.

Based on Eq. 24, the training data $Y(X_i, t_{ij})(i = 1, \dots, N, j = 1, \dots, N_i)$ is generated as follows:

$$Y(X_i, t_{ij}) = \sum_{n=1}^{N} \sum_{l=1}^{L} k_G(X_i, X_n)\theta_{nl} k_T(t_{ij}, t_l) + \epsilon_{(i,ij)},$$

$$\epsilon_{(i,ij)} \overset{i.i.d}{\sim} N(0, \ \sigma^2), \quad (i = 1, \dots, N, \ j = 1, \dots, N_i). \tag{25}$$

Let $\boldsymbol{T}_i$ and $\boldsymbol{g}_i$ $(i = 1, \dots, N)$ be $N_i \times L$ matrices and $N$-dimensional vector, respectively, expressed as follows:

$$\boldsymbol{T}_i = \begin{pmatrix} k_T(t_{i1}, t_1) & \cdots & k_T(t_{i1}, t_t) & \cdots & k_T(t_{i1}, t_L) \\ \vdots & \ddots & & & \vdots \\ k_T(t_{ij}, t_1) & & k_T(t_{ij}, t_t) & & k_T(t_{ij}, t_L) \\ \vdots & & & \ddots & \vdots \\ k_T(t_{iN_i}, t_1) & \cdots & k_T(t_{iN_i}, t_t) & \cdots & k_T(t_{iN_i}, t_L) \end{pmatrix},$$

$$\boldsymbol{g}_i = (k_G(X_i, X_1), \cdots, k_G(X_i, X_N))^\top. \tag{26}$$

Let $\boldsymbol{h}_i$ $(i = 1, \dots, N)$ be $N_i \times NL$ matrices given as $\boldsymbol{h}_i = \boldsymbol{g}_i^\top \otimes \boldsymbol{T}_i$ and denote $\sum_{i=1}^{N} N_i = S$. Using these matrices, we define the $S \times NL$ matrix $\boldsymbol{H}$ as follows:

$$\boldsymbol{H} = \begin{pmatrix} \boldsymbol{h}_1 \\ \vdots \\ \boldsymbol{h}_i \\ \vdots \\ \boldsymbol{h}_N \end{pmatrix}. \tag{27}$$

Using $H$, Eq. 25 can be rewritten as

$$\boldsymbol{y} = \boldsymbol{H} \cdot \boldsymbol{\theta} + \boldsymbol{\epsilon}, \tag{28}$$

where $\boldsymbol{y} = (Y(X_1, t_{11}), \cdots, Y(X_1, t_{1N_1}), \cdots, Y(X_N, t_{N1}), \cdots, Y(X_N, t_{NN_N}))^\top \in \mathbb{R}^S$,
$\boldsymbol{\theta} = (\theta_{11}, \cdots, \theta_{1L}, \theta_{21}, \cdots, \theta_{2L}, \cdots, \theta_{N1}, \cdots, \theta_{NL})^\top \in \mathbb{R}^{NL}$, and $\boldsymbol{\epsilon}$ is an $S$-dimensional random vector whose entries are independently and identically distributed according to a normal distribution $N(0, \ \sigma^2)$.

By summarizing Eq. 28, the likelihood of KRSFD $p(\boldsymbol{y}|\boldsymbol{\theta}, \sigma^2)$ is expressed as

$$p(\boldsymbol{y}|\boldsymbol{\theta}, \sigma^2) = N(\boldsymbol{H}\boldsymbol{\theta}, \ \sigma^2 \boldsymbol{I}_{NL}). \tag{29}$$

The prior distributions of the model parameters $\boldsymbol{\theta}$ and variance $\sigma^2$ are given as follows:

$$p(\boldsymbol{\theta}) = N\big(\boldsymbol{0}, \ (2\lambda' \boldsymbol{I}_{NL})^{-1}\big),$$

$$p(\sigma^2) = IG(\alpha, \beta). \tag{30}$$

From Eqs. 29 and 30, the full conditional posterior distributions are derived as follows:

$$p(\boldsymbol{\theta}|\boldsymbol{y}, \sigma^2) = N(\boldsymbol{\mu}_\theta, \ \boldsymbol{\Sigma}_\theta),$$

$$\boldsymbol{\mu}_\theta = (\boldsymbol{H}^\top \boldsymbol{H} + \lambda \boldsymbol{I}_{NL})^{-1} \boldsymbol{H}^\top \boldsymbol{y},$$

$$\boldsymbol{\Sigma}_\theta = \sigma^2 \cdot (\boldsymbol{H}^\top \boldsymbol{H} + \lambda \boldsymbol{I}_{NL})^{-1},$$

$$p(\sigma^2|\boldsymbol{y}, \boldsymbol{\theta}) = IG(\alpha + S/2, \ \beta + 1/2\|\boldsymbol{y} - \boldsymbol{H}\boldsymbol{\theta}\|_2^2). \tag{31}$$

The hyperparameter $\lambda'$ in Eq. 30 is substituted by $\lambda' = \lambda/2\sigma^2$ in Eq. 31.

Similarly to KRFD, the variance $\sigma^2$ of KRSFD is estimated as the MAP solution of $\sigma^2$, given the MAP solution of $\boldsymbol{\theta}$, $\boldsymbol{\theta}_{MAP} = \underset{\boldsymbol{\theta}}{\arg\max} \, p(\boldsymbol{\theta}|\boldsymbol{y}, \sigma^2)$, as follows:

$$\sigma^2_{MAP} = \underset{\boldsymbol{\sigma^2}}{\arg\max} \, p(\sigma^2|\boldsymbol{y}, \boldsymbol{\theta}_{MAP}) = \frac{2\beta + \|\boldsymbol{y} - \boldsymbol{H}\boldsymbol{\theta}_{MAP}\|_2^2}{2\alpha + 2 + S}, \tag{32}$$

where $\boldsymbol{\theta}_{MAP} = \boldsymbol{\mu}_\theta = (\boldsymbol{H}^\top \boldsymbol{H} + \lambda \boldsymbol{I}_{NL})^{-1} \boldsymbol{H}^\top \boldsymbol{y}$.

Similarly to KRFD, the prediction distribution of KRSFD for a new input $(X_{new}, t_{new})$ can be written as follows:

$$
\begin{aligned}
p\big(\hat{Y}(X_{new}, t_{new}) | \boldsymbol{y}, \sigma^2_{MAP}\big) &= N(\mu_{pred}, \; \sigma^2_{pred}), \\
\mu_{pred} &= (\boldsymbol{g}_{new} \otimes \boldsymbol{t}_{new})^\top (\boldsymbol{H}^\top \boldsymbol{H} + \lambda \boldsymbol{I}_{NL})^{-1} \boldsymbol{H}^\top \boldsymbol{y}, \\
\sigma^2_{pred} &= \sigma^2_{MAP} \cdot (\boldsymbol{g}_{new} \otimes \boldsymbol{t}_{new})^\top (\boldsymbol{H}^\top \boldsymbol{H} + \lambda \boldsymbol{I}_{NL})^{-1} (\boldsymbol{g}_{new} \otimes \boldsymbol{t}_{new}),
\end{aligned}
\tag{33}
$$

where $\boldsymbol{g}_{new} = (k_G(X_{new}, X_1), \ldots, k_G(X_{new}, X_N))^\top$ and $\boldsymbol{t}_{new} = (k_T(t_{new}, t_1), \ldots, k_T(t_{new}, t_L))^\top$.

In Eq. 33, we must calculate the inverse of the $NL \times NL$ matrix $(\boldsymbol{H}^\top \boldsymbol{H} + \lambda \boldsymbol{I}_{NL})$. The memory requirements and computational complexity are $\mathcal{O}(N^2 L^2)$ and $\mathcal{O}(N^3 L^3)$, respectively, which become challenging to manage unless both $N$ and $L$ are kept low. Therefore, as described later, our implementation adapts several techniques, which improve the scalability with respect to sample size.

In KRFD where the measurement point set on $t$ is the same across different $X$ and is set to be the kernel centers of $k_T(\cdot, \cdot)$, the computational complexity of $\boldsymbol{H}$ can be significantly reduced because the Kronecker product can decompose it as $\boldsymbol{H} = \boldsymbol{G} \otimes \boldsymbol{T}$. This allows the inverse matrices of $\boldsymbol{G}$ and $\boldsymbol{T}$ to be computed separately to derive the MAP solutions, as shown in Eqs. 18 and 19. Here, as shown in Eq. 18, the prior distribution defined in Eq. 10 yields the MAP solution concisely. In this form, the computation of the inverse matrix is separated for $(\boldsymbol{G} + \lambda_G \boldsymbol{I}_N)$ and $(\boldsymbol{T} + \lambda_T \boldsymbol{I}_L)$. Furthermore, hyperparameters $\lambda_G$ and $\lambda_T$ are assigned to $\boldsymbol{G}$ and $\boldsymbol{T}$, separately. The above results demonstrate that KRFD has significantly lower memory requirements and computational complexity than KRSFD, with KRFD requiring $\max\big(\mathcal{O}(N^2), \mathcal{O}(L^2)\big)$ memory and having a computational complexity of $\max\big(\mathcal{O}(N^3), \mathcal{O}(L^3)\big)$.

To cope with the scalability problem, we employ a truncated kernel to reduce memory requirements. The sparse Gram matrices calculated using the truncated kernels are efficiently handled using the sparse matrix module from the SciPy Python library Virtanen et al. (2020). In addition, to save computational costs, the conjugate gradient (CG) method Hestenes et al. (1952) is employed to calculate the mean of the parameter posterior distribution $\boldsymbol{\mu}_\theta$ in Eq. 31. If only the prediction mean is needed, rather than the entire prediction distribution of the function, it is sufficient to calculate $\boldsymbol{\mu}_\theta$ only. In the CG method, $\boldsymbol{\mu}_\theta$ can be derived without calculating $(\boldsymbol{H}^\top \boldsymbol{H} + \lambda \boldsymbol{I}_{NL})^{-1}$ by solving the system of linear equations represented as $(\boldsymbol{H}^\top \boldsymbol{H} + \lambda \boldsymbol{I}_{NL}) \boldsymbol{x} = \boldsymbol{H}^\top \boldsymbol{y}$ for the vector $\boldsymbol{x}$. We iterated the CG algorithm until either the mean square error of the residual vector $\boldsymbol{r} = (\boldsymbol{H}^\top \boldsymbol{H} + \lambda \boldsymbol{I}_{NL}) \boldsymbol{x} - \boldsymbol{H}^\top \boldsymbol{y}$ fell below 0.001 or the number of iterations reached 500.

To compute the standard deviation of the predicted functions and generate samples from the predictive distribution, it is necessary to obtain the covariance matrix of the posterior distribution, $\boldsymbol{\Sigma}_\theta$, as shown in Eq. 31. This involves performing the matrix inversion of $(\boldsymbol{H}^\top \boldsymbol{H} + \lambda \boldsymbol{I}_{NL})$. To reduce the computational complexity for this calculation, we employ the incomplete lower-upper (LU) decomposition Saad (1994), which provides an approximate inverse of $(\boldsymbol{H}^\top \boldsymbol{H} + \lambda \boldsymbol{I}_{NL})$. We use the SciPy Python library Virtanen et al. (2020) to perform the incomplete LU decomposition of a sparse matrix.

## 2.4 Representer theorem

The KRFD model is naturally formulated from the perspective of an RKHS. An input pair $(X_i, t_j)$ represents a point in the $\mathbb{R}^{p+q}$ space. Let $\mathcal{H}_\gamma$ be an RKHS composed of functions on $\mathbb{R}^{p+q}$, with kernel $\gamma : \mathbb{R}^{p+q} \times \mathbb{R}^{p+q} \to \mathbb{R}$. Here, we construct a prediction model using the inner product of the input pair $(X_i, t_j)$ mapped onto the feature space $\mathcal{H}_\gamma$ and an arbitrary function $f(X, t)$ in $\mathcal{H}_\gamma$. The objective function is then given as

$$
\min_{f(X,t) \in \mathcal{H}_\gamma} \sum_{i=1}^N \sum_{j=1}^L \| Y(X_i, t_j) - \langle f(X,t), \gamma((X_i, t_j), (X, t)) \rangle_{\mathcal{H}_\gamma} \|^2 + \lambda \| f(X,t) \|^2_{\mathcal{H}_\gamma},
\tag{34}
$$

where $\langle \cdot, \cdot \rangle_{\mathcal{H}_\gamma}$ denotes the inner product in $\mathcal{H}_\gamma$. Suppose that the kernel $\gamma$ in Eq. 34 is separable, $\gamma((X_1, t_1), (X_2, t_2)) = k_G(X_1, X_2) \cdot k_T(t_1, t_2)$, where $(X_1, t_1), (X_2, t_2) \in \mathbb{R}^{p+q}, X_1, X_2 \in \mathbb{R}^p$, and $t_1, t_2 \in \mathbb{R}^q$, and $k_G$ and $k_T$ are positive definite kernels defined in the $\mathbb{R}^p$ and $\mathbb{R}^q$ spaces, respectively, leading to $\langle f(X,t), \gamma((X_i, t_j), (X, t)) \rangle_{\mathcal{H}_\gamma} = \langle f(X,t), k_G(X_i, X) k_T(t_j, t) \rangle_{\mathcal{H}_\gamma}$.

We add regularization terms to Eq. 34 that impose separate constraints on $X$ and $t$ with respect to the complexity of the KRFD. Let $\mathcal{H}_G$ and $\mathcal{H}_T$ be the RKHSs defined by the functions in spaces $\mathbb{R}^p$ and $\mathbb{R}^q$ and kernels $k_G$ and $k_T$, respectively. If $f(X, t_j)$ and $f(X_i, t)$ are the elements in $\mathcal{H}_G$ and $\mathcal{H}_T$, respectively, we can add regularization terms based on the norms of $\mathcal{H}_G$ and $\mathcal{H}_T$ as follows:

$$\min_{f(X,t)\in\mathcal{H}_\gamma} \sum_{i=1}^{N} \sum_{j=1}^{L} \|Y(X_i, t_j) - \langle f(X,t), \gamma((X_i, t_j), (X, t))\rangle_{\mathcal{H}_\gamma}\|^2$$

$$+\lambda_a \sum_{j=1}^{L} \|f(X, t_j)\|_{\mathcal{H}_G}^2 + \lambda_b \sum_{i=1}^{N} \|f(X_i, t)\|_{\mathcal{H}_T}^2 + \lambda_c \|f(X, t)\|_{\mathcal{H}_\gamma}^2, \tag{35}$$

where $\lambda_a$, $\lambda_b$, and $\lambda_c$ are hyperparameters that control the influence of the regularization terms.

According to the representer theorem Schölkopf et al. (2001), $f(X, t) \in \mathcal{H}_\gamma$ can be represented as $f(X, t) = \sum_{n=1}^{N} \sum_{l=1}^{L} \theta_{nl} k_G(X_n, X) k_T(t_l, t)$ to minimize Eq. 35 using the given training set. Substituting this into Eq. 35 yields the following exppression:

$$\min_{\boldsymbol{\Theta}} \|\boldsymbol{Y} - \boldsymbol{G\Theta T}\|_F^2 + \lambda_a \|\boldsymbol{G}^{1/2}\boldsymbol{\Theta T}\|_F^2 + \lambda_b \|\boldsymbol{G\Theta T}^{1/2}\|_F^2 + \lambda_c \|\boldsymbol{G}^{1/2}\boldsymbol{\Theta T}^{1/2}\|_F^2, \tag{36}$$

where $\boldsymbol{Y}$ and $\boldsymbol{\Theta}$ are matrices with elements obtained from $\boldsymbol{Y}_{ij} = Y(X_i, t_j)$ and $\boldsymbol{\Theta}_{ij} = \theta_{ij}$, respectively, and $\|\cdot\|_F^2$ denotes the Frobenius norm. Each term in Eq. 36 is derived as follows:

$$\langle f(X,t), \gamma((X_i, t_j), (X, t))\rangle_{\mathcal{H}_\gamma} = \langle f(X,t), k_G(X_i, X)k_T(t_j, t)\rangle_{\mathcal{H}_\gamma}$$

$$= \langle \sum_{n=1}^{N} \sum_{l=1}^{L} \theta_{nl} k_G(X_n, X)k_T(t_l, t), k_G(X_i, X)k_T(t_j, t)\rangle_{\mathcal{H}_\gamma}$$

$$= \sum_{n=1}^{N} \sum_{l=1}^{L} k_G(X_i, X_n)\theta_{nl}k_T(t_l, t_j) = (\boldsymbol{G\Theta T})_{ij},$$

$$\|f(X,t)\|_{\mathcal{H}_\gamma}^2 = \langle f(X,t), f(X,t)\rangle_{\mathcal{H}_\gamma}$$

$$= \langle \sum_{n=1}^{N} \sum_{l=1}^{L} \theta_{nl} k_G(X_n, X)k_T(t_l, t), \sum_{n=1}^{N} \sum_{l=1}^{L} \theta_{nl} k_G(X_n, X)k_T(t_l, t)\rangle_{\mathcal{H}_\gamma}$$

$$= \text{tr}(\boldsymbol{G\Theta T\Theta}^\top) = \|\boldsymbol{G}^{1/2}\boldsymbol{\Theta T}^{1/2}\|_F^2,$$

$$\sum_{j=1}^{L} \|f(X, t_j)\|_{\mathcal{H}_G}^2 = \sum_{j=1}^{L} \langle (\boldsymbol{\Theta t}_j)^\top \boldsymbol{\Phi}_G(X), (\boldsymbol{\Theta t}_j)^\top \boldsymbol{\Phi}_G(X)\rangle_{\mathcal{H}_G} = \text{tr}(\boldsymbol{G\Theta T}^2\boldsymbol{\Theta}^\top) = \|\boldsymbol{G}^{1/2}\boldsymbol{\Theta T}\|_F^2,$$

$$\sum_{i=1}^{N} \|f(X_i, t)\|_{\mathcal{H}_T}^2 = \sum_{i=1}^{N} \langle \boldsymbol{g}_i^\top \boldsymbol{\Theta}\boldsymbol{\Phi}_T(t), \boldsymbol{g}_i^\top \boldsymbol{\Theta}\boldsymbol{\Phi}_T(t)\rangle_{\mathcal{H}_T} = \text{tr}(\boldsymbol{G}^2\boldsymbol{\Theta T\Theta}^\top) = \|\boldsymbol{G\Theta T}^{1/2}\|_F^2, \tag{37}$$

where $\boldsymbol{\Phi}_G(X) = (k_G(X_1, X), \ldots, k_G(X_N, X))^\top$ and $\boldsymbol{\Phi}_T(t) = (k_T(t_1, t), \cdots, k_T(t_L, t))^\top$, and $\boldsymbol{g}_i$ and $\boldsymbol{t}_j$ represent the $i$-th and $j$-th column vectors of $\boldsymbol{G}$ and $\boldsymbol{T}$, respectively.

If $\lambda_a = \lambda_G$, $\lambda_b = \lambda_T$, and $\lambda_c = \lambda_G\lambda_T$, then the optimal solution to Eq. 36 is given by

$$\hat{\boldsymbol{\Theta}} = (\boldsymbol{G} + \lambda_G\boldsymbol{I}_N)^{-1}\boldsymbol{Y}(\boldsymbol{T} + \lambda_T\boldsymbol{I}_L)^{-1}. \tag{38}$$

Let $\text{vec}(\cdot)$ be the vectorization operator. Here, if $\boldsymbol{\theta} = \text{vec}(\boldsymbol{\Theta}^\top)$, $\boldsymbol{y} = \text{vec}(\boldsymbol{Y}^\top)$, and $\text{vec}(\boldsymbol{ABC}) = \boldsymbol{C}^\top \otimes \boldsymbol{A} \cdot \text{vec}(\boldsymbol{B})$, Eq. 38 can be rewritten as

$$\hat{\boldsymbol{\theta}} = (\boldsymbol{G} + \lambda_G\boldsymbol{I}_N)^{-1} \otimes (\boldsymbol{T} + \lambda_T\boldsymbol{I}_L)^{-1}\boldsymbol{y}. \tag{39}$$

Comparing Eqs. 39 and 18, the optimal solution $\hat{\boldsymbol{\theta}}$ is identical to the MAP solution in the Bayesian KRFD, except for the term $-\boldsymbol{M} \otimes \boldsymbol{1} \cdot \boldsymbol{c}_{MAP}$. If we include the $t$-independent term $\mu(X)$ in the model, the following modifications are made to Eq. 35; the term $\langle f'(X), k_M(X_i, X)\rangle_{\mathcal{H}_M}$ is added to the model, and

$\lambda_M \sum_{j=1}^{L} \|f'(X)\|^2_{\mathcal{H}_M}$ is added to the regularization term. Here, $\mathcal{H}_M$ is an RKHS with the positive definite kernel $k_M(X, X)$, and $f'(X)$ is an arbitrary element in $\mathcal{H}_M$. By applying the representer theorem, $f'(X)$ can be expresssed as $f'(X) = \sum_{m=1}^{N} c_m k_M(X_m, X)$. Solving the objective function under this form yields the optimal solution $\hat{\boldsymbol{\theta}}$ and $\hat{\boldsymbol{c}}$, which can be derived in the same way as the simultaneous MAP solution in Eqs. 18 and 19. This shows that the form and solution of the KRFD are naturally derived from an optimization problem in the RKHS, enabling the interpretation of the KRFD from an RKHS perspective. For example, defining the prior distribution of $\boldsymbol{\theta}$ as shown in Eq. 10, where $\lambda_a = \lambda_G/2\sigma^2$, $\lambda_b = \lambda_T/2\sigma^2$, and $\lambda_c = \lambda_G \lambda_T/2\sigma^2$, is equivalent to adding the regularization terms as $\lambda_G \sum_{j=1}^{L} \|f(X, t_j)\|^2_{\mathcal{H}_G} + \lambda_T \sum_{i=1}^{N} \|f(X_i, t)\|^2_{\mathcal{H}_T} + \lambda_G \lambda_T \|f(X, t)\|^2_{\mathcal{H}_\gamma}$. We can interpret Eq. 10 as the regularizer that penalizes the smoothness of the KRFD separately on the $X$ and $t$ spaces. Their influences can be controlled independently by the hyperparameters $\lambda_G$ and $\lambda_T$.

## 2.5 Relations to multitask learning

Deriving KRFD from an RKHS perspective reveals its connection to the MTL framework proposed by Ciliberto et al. (2015). The MTL framework generally leverages task similarities to enhance learning efficiency and prediction accuracy across multiple tasks. Ciliberto et al. proposed an MTL model based on an RKHS of vector-valued functions, where the use of separable kernels allows task similarities and input similarities to be encoded separately. Notably, the KRFD's objective function in Eq. 36, where $\lambda_a = \lambda_b = 0$, is equivalent to the objective function of Ciliberto et al. (2015) when the task similarity matrix is known and the loss function is the $\ell_2$ norm. In this context, the task similarity matrix corresponds to the matrix $\boldsymbol{T}$ in KRFD, as each task in MTL can be interpreted as corresponding to a measurement point in the functional output prediction.

Similarly, the MTL model proposed by Bonilla et al. (2007) has a close relationship with the KRFD model. Bonilla's model is a Gaussian process (GP) model in which task similarities are encoded through a covariance matrix. In this model, a latent function $f_t(X)$ is assumed for each task $t$, which maps the input $X$ to a real-valued output. The latent functions are governed by a GP prior, with the covariance matrix given by $\mathrm{Cov}(f_t(X), f_{t'}(X')) = k_T(t, t')\, k_X(X, X')$, where $k_T(t, t')$ denotes the kernel function capturing task similarities and $k_X(X, X')$ denotes the kernel function modeling input similarities. As evident from this, Bonilla's model represents an MTL framework that employs separable kernels and is highly similar to KRFD.

KRFD can be regarded as a generalization of the MTL methods, with several key advantages. One of its distinct features is its ability to handle cases where the set of measurement points varies continuously across inputs, a capability traditional MTL models lack. In addition, by incorpolating the regularization terms in Eq. 36, our method significantly reduces computational complexity when all functional output values are observed at the same set of measurement points. Specifically, the complexity decreases from $\mathcal{O}(N^3 L^3)$ to $\max\big(\mathcal{O}(N^3), \mathcal{O}(L^3)\big)$ without requiring special matrix transformations or approximations. In terms of uncertainty quantification, a notable distinction exists between KRFD and related models. While KRFD adopts a Bayesian perspective to analytically quantify predictive uncertainty—an aspect not provided by Ciliberto's model—its uncertainty is derived from the parameter posterior rather than the function posterior, as in GPs. Concequently, Bonilla's GP-based model provides more rigorous uncertainty quantification. However, KRFD offers greater flexibility in controlling model complexity and stability through explicit regularization terms. In addition, its transparent and straightforward formulation enhances interpretability, making it a practical and robust choice for various applications.

# 3 Results

This study applied the KRFD model to three examples, which are detailed in the following subsections.

## 3.1 Application to dense artificial data

To evaluate the predictive ability of KRFD model, we generated artificial data using the following formula.

$$y(t) = a\sin(bt + c) + dt + e + \epsilon, \ \epsilon \overset{i.i.d}{\sim} N(0, \ \sigma^2) \tag{40}$$

The above function is a composite of a sine wave and a straight line, and the five coefficients $a, b, c, d$, and $e$ determine the form of the function. The coefficients $a, b, c, d$, and $e$ can be interpreted as amplitude, frequency, phase, slope, and intercept, respectively. $\epsilon$ is the observation noise for each observation point. Here, the covariates for predicting the functional output $y(t)$ are denoted by a vector of the five coefficients, $X_i = (a_i, b_i, c_i, d_i, e_i)^\top$ ($i = 1, \ldots, N$). $N$ was set to 1,000. All function data $Y(X_i, t)$ were observed at 51 measurement points, evenly distributed between 0 and 2. The values of the coefficients $a$ and $b$ were randomly sampled from the uniform distribution with support $[1, 5]$ ($= U(1, 5)$), the values of $c$ were sampled from $U(0, 3)$, the values of $d$ were sampled from $U(-2, 2)$, and the values of $e$ were sampled from $U(-3, 3)$. $\sigma$ was set to be 0.2, and the values of $\epsilon$ were added to each observation point. The resulting artificial data were summarized in a $1000 \times 51$ observation matrix $\boldsymbol{Y}$, a $1000 \times 5$ covariates matrix $\boldsymbol{X}$, and a 51-dimensional measurement point vector $\boldsymbol{t}$. Training and test inputs were randomly split in a 3:1 ratio. To calculate the standard deviation of the predictions, we prepared five randomly split training-test datasets.

For comparison, we applied the FLM and KRR models. FLM is the most basic FSR and is represented by a linear combination of basis functions on $t$-space and the covariates $X \in \mathbb{R}^p$, as shown in the following equation.

$$Y(X, t) = \beta_0(t) + \sum_{j=1}^{p} x_j \beta_j(t), \ X = (x_1, \ldots, x_p)^\top. \tag{41}$$

In this study, the basis function $\beta_j(t)$ ($j = 0, \ldots, P$) in Eq. 41 was modeled as $\beta_j(t) = \sum_{i=1}^{L} k_T(t_i, t)\theta_i^j$, which is a linear combination of the trainable parameters $\{\theta_1^j, \ldots, \theta_L^j\}$ and the kernel functions on the measurement point set $\{t_1, \ldots, t_L\}$. Therefore, the total number of training parameters for FLM was $(P + 1) \times L$. The details of FLM are summarized in Appendix A. KRRs involve fitting a single-task KRR model to the observed values at each measurement point in $t$-space. This can be formulated as follows:

$$Y(X, t_j) = \sum_{i=1}^{N} k_G(X_i, X)\theta_i^j \ \ (j = 1, \ldots, L) \tag{42}$$

For all the three models, the kernel function was chosen to be either a Gaussian: $k(X, X') = \exp(-\|X - X'\|_2^2 / 2\sigma_{kernel}^2)$ or a Laplacian: $k(X, X') = \exp(-|X - X'| / \sigma_{kernel})$, where $\sigma_{kernel}$ is the scale parameter for the kernel functions. The model's hyperparameters were Bayesian optimized using the Optuna Python library Akiba et al. (2019) with five-fold cross-validation of a training dataset. To reduce computational cost, hyperparameter optimization was performed using the training portion of the first training-test dataset from the five randomly split datasets. Using the selected hyperparameters, the models were trained on the training portions of all five datasets, and the prediction performances were evaluated on the corresponding test datasets. The mean and standard deviation of the prediction performances were then calculated.

For KRFD, the hyperparameters consist of eight variables, $\{\lambda_G, \lambda_T, \lambda_M, \sigma_G, \sigma_T, \sigma_M, k_X, k_T\}$. $\lambda_G$, $\lambda_T$ and $\lambda_M$ determine the shape of the prior distribution of the trainable parameters (equivalent to those of Eq. 23), whereas $\sigma_G$, $\sigma_T$ and $\sigma_M$ are the scale parameters for the kernel functions $k_G(X, X')$, $k_T(t, t')$, and $k_M(X, X')$, respectively. The parameters $k_X$ and $k_T$ specify the type of kernel functions defined on $X$-space (i.e., $k_G(X, X')$ and $k_M(X, X')$) and $t$-space (i.e., $k_T(t, t')$), respectively. The solution space was defined as $\lambda_G, \lambda_T, \lambda_M \in [10^{-6}, 1]$, $\sigma_G, \sigma_T, \sigma_M \in [0.1, 100]$, and $k_X, k_T \in \{Gaussian, Laplacian\}$. For $\lambda_G, \lambda_T, \lambda_M, \sigma_G, \sigma_T$, and $\sigma_M$, the values were sampled from the logarithmic domain. The number of trials for hyperparameter optimization was set to 300. The best hyperparameters obtained were: $\lambda_G = 1.725 \times 10^{-4}, \lambda_T = 0.052, \lambda_M = 1.500 \times 10^{-5}, \sigma_G = 1.963, \sigma_T = 0.466, \sigma_M = 13.026, k_X = Gaussian, k_T = Gaussian$.

For FLM, the hyperparameters comprise three variables $\{\lambda, \sigma, k_T\}$, where $\lambda$ controls the regularization strength (see Appendix A for details), $\sigma$ is the scale parameter for $k_T(t, t')$ and $k_T$ is the type of kernel function defined in $t$-space. The solution space was defined as $\lambda \in [10^{-6}, 1]$, $\sigma \in [0.1, 100]$, and $k_T \in \{Gaussian, Laplacian\}$. The values of $\lambda$ and $\sigma$ were sampled from the logarithmic domain. The number of trials was set to 300. The best hyperparameters were determined to be $\lambda = 0.044, \sigma = 0.532, k_T = Gaussian$.

Single-task KRR models were trained independently, for each $t_i$ ($i = 1, \ldots, 51$). Therefore, hyperparameter optimization was performed independently for each KRR (the number of trials was set to 30 for each KRR

model). The KRRs were trained using the scikit-learn Python library Pedregosa et al. (2011). Thus, the hyperparameters follow those used in the scikit-learn library. The hyperparameters comprise three variables $\{\alpha, \gamma, k_X\}$, where $\alpha$ is the regularization strength, $\gamma$ is an inverse scale parameter equivalent to $1/2\sigma_{kernel}^2$ for a Gaussian and $1/\sigma_{kernel}$ for Laplacian kernel, and the $k_X$ is the type of kernel function defined in $X$-space. The solution space was defined as $\alpha \in [10^{-6}, 1]$, $\gamma \in [10^{-6}, 1]$, and $k_X \in \{Gaussian, Laplacian\}$. The values of $\alpha$ and $\gamma$ were sampled from the logarithmic domain. The best hyperparamers for the 51 KRR models are summarized in Figure B1.

Table 1 reports the mean absolute error (MAE), root-mean-square error (RMSE), $R^2$, and mean correlation coefficient (mean R) with respect to the test sets. The mean R was calculated by first determining the correlation coefficient (R) between the predicted and observed values for each input and then averaging the R values across all inputs in the test dataset. MAE, RMSE, and $R^2$ were calculated between the combined predicted and observed values across all inputs in the test dataset. The performance metrics were averaged over five trials, and the numbers in parentheses represent the standard deviations. The parity plots of the predicted and observed values are shown in the top row of Figure 2. The bottom row of Figure 2 shows the histograms of the R values for each input. Examples of individual function predictions on the test samples by each model are shown in Figure 3. Because KRFD can provide the prediction distribution for a new input $(X_{new}, t_{new})$, as shown in Eq. 23, the functional prediction results with a $\pm 1$ band of the standard deviation and the results of functional sampling from the prediction distribution are also shown in Figure 3 for KRFD. Since KRRs and FLM do not provide prediction distributions; therefore, only function prediction results are shown here. As shown in Table 1, KRFD outperformed the other models in all performance metrics. Because KRRs consist of a set of independent predictive models corresponding to each point of $t$-space, as shown in Figure 3, the function prediction results for KRRs showed a greater variability of predictions with respect to changes in $t$ compared with KRFD. This variability may have made it difficult for KRRs to accurately capture the underlying smoothness of the true data, leading to poorer prediction performance. The performance of FLM was much worse than that of the KRFD and KRR models. This may be because FLM is a linear model with respect to $X$ and may not have sufficient expressive power to represent the data characteristics; for example, in Eq. 40, the relationship between $X$ and function $Y(X, t)$ is inherently nonlinear.

Table 1: Model performance for the dense artificial data.

| Models | MAE | RMSE | $R^2$ | mean R |
|--------|-----|------|-------|--------|
| KRFD | 0.216 ($\pm$0.005) | 0.301 ($\pm$0.014) | 0.991 ($\pm$0.001) | 0.977 ($\pm$0.002) |
| KRRs | 0.278 ($\pm$0.009) | 0.404 ($\pm$0.025) | 0.983 ($\pm$0.002) | 0.966 ($\pm$0.004) |
| FLM | 1.278 ($\pm$0.036) | 1.663 ($\pm$0.047) | 0.718 ($\pm$0.018) | 0.723 ($\pm$0.018) |

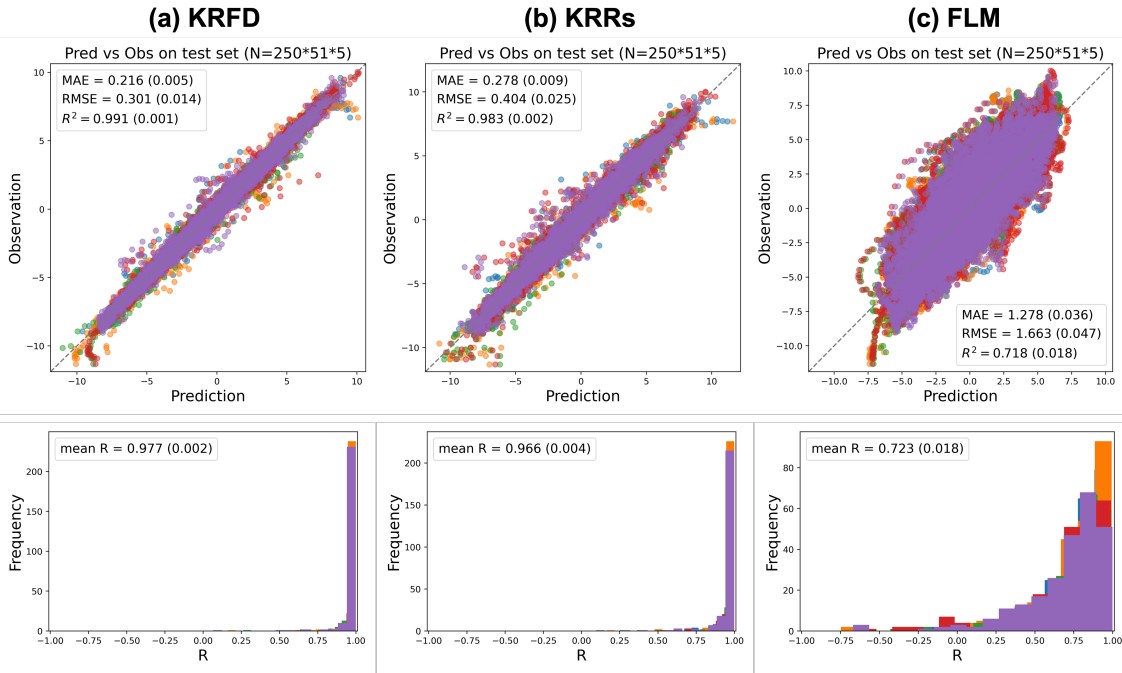

Figure 2: Prediction results for the test samples of the dense artificial data using the (a) KRFD, (b) KRRs, and (c) FLM models. The scatter plots and histograms are color-coded in five different colors corresponding to the five independent data splitting.

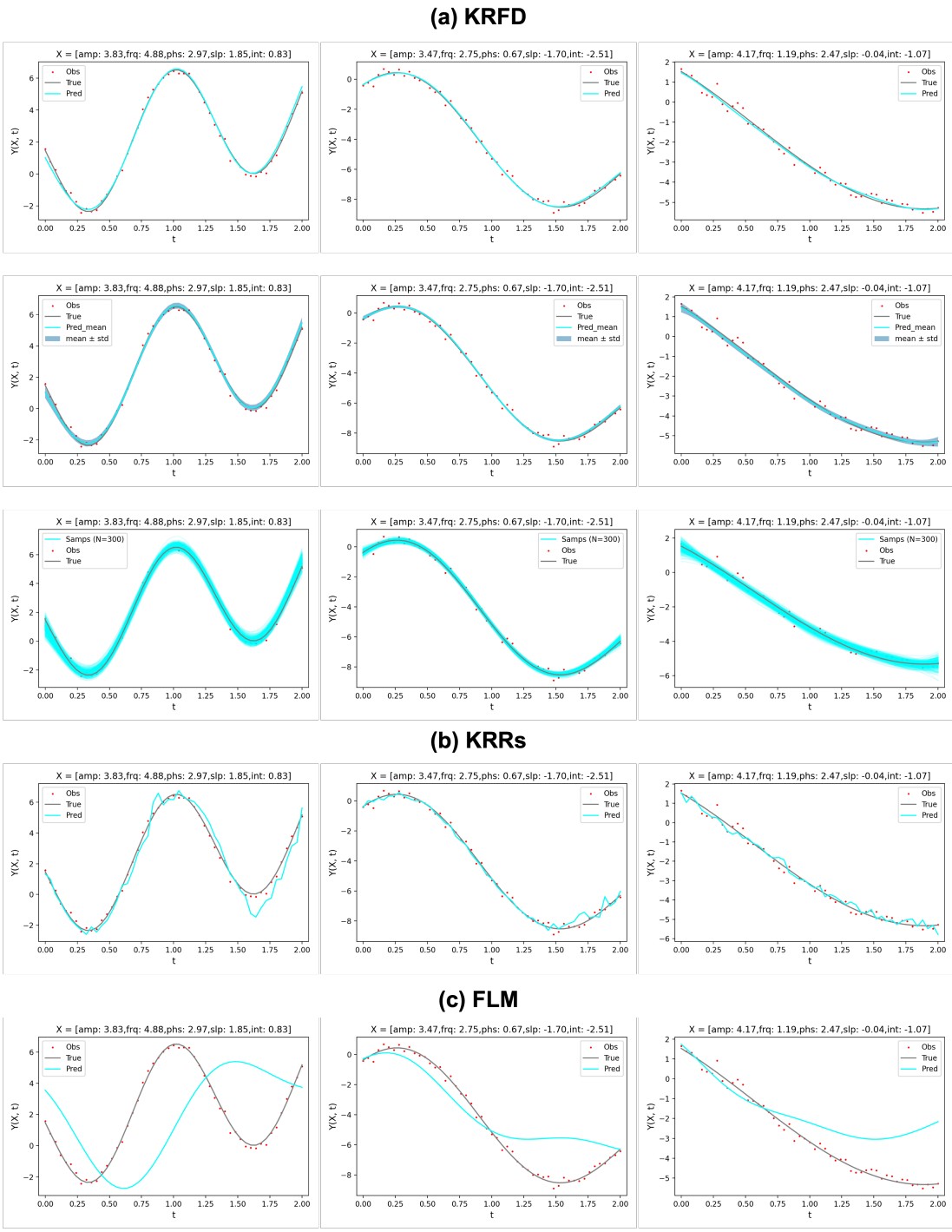

Figure 3: Functional prediction results for the test samples. The gray line shows the true data before adding observation noise, the red dots show the observed data points, and the light blue line shows the functional predictions. The title of each figure indicates the actual amplitude, frequency, phase, slope, and intercept values for that input. The first through third rows show the predictions using the KRFD model. The first row shows only the mean of the predictive distribution, the second row shows the predicted mean ±1 standard deviation (blue band) of the predictive distribution, and the third row shows the results of sampling the function 300 times from the predictive distribution. The fourth and fifth rows show the predicted results for KRRs and FLM, respectively.

## 3.2 Application to sparse artificial data

We generated sparse artificial data using Eq. 40 to test the predictive ability of KRSFD. The coefficients $a$, $b$, $c$, $d$, and $e$ comprising the covariates $X$ were generated by randomly sampling 1,000 times from $U(1,5)$ for $a$ and $b$, $U(0,3)$ for $c$, $U(-2,2)$ for $d$, and $U(-3,3)$ for $d$. The sparse measurement points $t_{ij}(i = 1,\ldots,N, j = 1,\ldots,N_i)$ in $t$-space were generated by first randomly selecting the number of measurement points $N_i$ corresponding to each input $X_i$ from the integer set $\{2,3,\ldots,20\}$ and then randomly sampling $N_i$ instances from $U(0,2)$. Then, the observed data $Y(X_i, t_{ij})$ was generated using Eq. 40. The standard deviation of the observation noise $\sigma$ was set to 0.2. The total number of measurement points across the 1,000 inputs, $M = \sum_{i=1}^{1,000} N_i$, was 10,962. Therefore, the resulting sparse artificial data are summarized in a $1000 \times 5$ covariates matrix $\boldsymbol{X}$, a 10,962-dimensional measurement point vector $\boldsymbol{t}$, and a 10,962-dimensional observation vector $\boldsymbol{y}$. The covariates matrix $\boldsymbol{X}$ was exactly the same as the one used in Section 3.1.

As discussed in the Methods section and Appendix A, the KRSFD and FLM models require a kernel center set $t_1,\ldots,t_L$ for $k_T(t,\cdot)$ apart from the training data. In this example, the kernel center set was set to 30 grid points evenly distributed between 0 and 2. The 1,000 inputs were randomly divided into training and testing inputs in a 3:1 ratio. To calculate the standard deviation of the predictions, we prepared five randomly split training-test datasets. In this example, the number of training data points changes for each training-test dataset because the number of observations varies from input to input. The number of training data points in the five datasets were 8,228, 8,289, 8,176, 8,204, and 8,093. In this example, the KRSFD results were compared with those of FLM. KRR modeling was not possible because measurement points in $t$-space varied from input to input.

The model hyperparameters were Bayesian optimized using Optuna with five-fold cross-validation. Hyperparameter optimization was performed using the training set of the first of the five randomly split training-test datasets. Using the selected hyperparameters, the models were trained on the training sets of the five datasets, and the prediction performances were evaluated on the corresponding test sets. The mean and standard deviation of the prediction performances were then calculated.

The KRSFD model has six hyperparameters: $\{\lambda, \sigma_G, \sigma_T, z_G, k_X, k_T\}$. $\lambda$ determines the shape of the prior distribution of the trainable parameters, equivalent to $\lambda$ in Eq. 31. $\sigma_G$ and $\sigma_T$ are the scale parameters of the kernels $k_G(X, X')$ and $k_T(t, t')$, respectively. $z_G$ represents the proportion of zeros in the Gram matrix generated from $k_G(X, X')$. $k_X$, and $k_T$ specify the types of kernel functions defined in $X$-space and $t$-space, respectively.

The proportion of zeros in the Gram matrix generated from $k_T(t, t')$ is denoted as $z_T$ and defined as: $z_T = (0.9 - z_G)/(1 - z_G)$. If the proportion of zeros in the matrix $\boldsymbol{H} = \boldsymbol{G} \otimes \boldsymbol{T}$ is defined as $z_H$, then the following equation holds: $z_H = z_G + z_T - z_G \cdot z_T$. The aforementioned formula for $z_T$ ensures that $z_H$ is set to 0.9, which adjusts the matrix $\boldsymbol{H}$ of KRSFD (shown in Eq. 27) such that approximately 90% of its elements are zero, regardless of the value of $z_G$.

The solution space for hyperparameter optimization was defined as $\lambda \in [10^{-6}, 1]$, $\sigma_G, \sigma_T \in [0.1, 100]$, $z_G \in [0.1, 0.9]$, and $k_G(X, X'), k_T(t, t') \in \{Gaussian, Laplacian\}$. The number of trials was set to 300. The optimal hyperparameter values were $\lambda = 0.024$, $\sigma_G = 1.249$, $\sigma_T = 0.173$, $z_G = 0.434$ ($z_T = 0.823$ for this $z_G$), $k_G = Gaussian$, and $k_T = Laplacian$.

The hyperparameter set of the FLM model, its solution space, and the number of trials for the hyperparameter optimization were the same as for the dense artificial data. The optimal hyperparameter values were $\lambda = 1.258 \times 10^{-6}$, $\sigma = 0.827$, and $k_T = Gaussian$.

Table 2 reports the mean and standard deviation of the MAE, RMSE, $R^2$, and mean R metrics for the five independent experiments with different data splitting. The parity plots of the predicted and observed values are shown in the top row of Figure 4. The bottom row of Figure 4 shows the histograms of the R values for each input. Examples of individual function predictions on the test samples by the KRSFD and FLM models are shown in Figure 5. Because KRSFD can provide the prediction distribution for a new input $(X_{new}, t_{new})$, as shown in Eq. 33, the functional prediction results with a $\pm 1$ band of the standard deviation and the results of functional sampling from the prediction distribution are also shown in Figure 5 for KRSFD.

As shown in Table 2, KRSFD outperformed FLM in all performance metrics. FLM did not have sufficient expressive power to represent the nonlinear system described by Eq. 40. The prediction performance of KRSFD for the sparse data was worse than that of KRFD for the dense data. Even in numerical experiments where the total amount of data for KRFD was reduced to approximately match that of KRSFD, the prediction performance of KRSFD remained inferior (see Appendix C). The inferior prediction performance of KRSFD compared with KRFD was attributed to the increased complexity of the estimation problem caused by sparse data, where the measurement points varied across inputs. As shown in Figure 5, the functional predictions using KRSFD are not as smooth as that using KRFD, likely because a truncated kernel is used in the KRSFD model. The optimal hyperparameter values in the KRSFD model were $z_G = 0.434$ and $z_T = 0.823$, which indicates that more detailed covariance information was needed in $X$-space than in $t$-space for accurate functional prediction. In addition to the functional predictions on the test samples, KRSFD can also be applied to interpolate sparse functional data. This corresponds to making function predictions on the training data, and the results are summarized in Figure 6. The middle panel in Figure 6 shows that KRSFD performs nonlinear interpolation very close to the true data from only four observation points by utilizing the covariance information in $X$-space.

Table 2: Model performance for the sparse artificial data.

| Models | MAE | RMSE | $R^2$ | mean R |
|---|---|---|---|---|
| KRSFD | 0.398 ($\pm$0.014) | 0.564 ($\pm$0.032) | 0.966 ($\pm$0.002) | 0.935 ($\pm$0.009) |
| FLM | 1.252 ($\pm$0.031) | 1.622 ($\pm$0.049) | 0.720 ($\pm$0.015) | 0.698 ($\pm$0.026) |

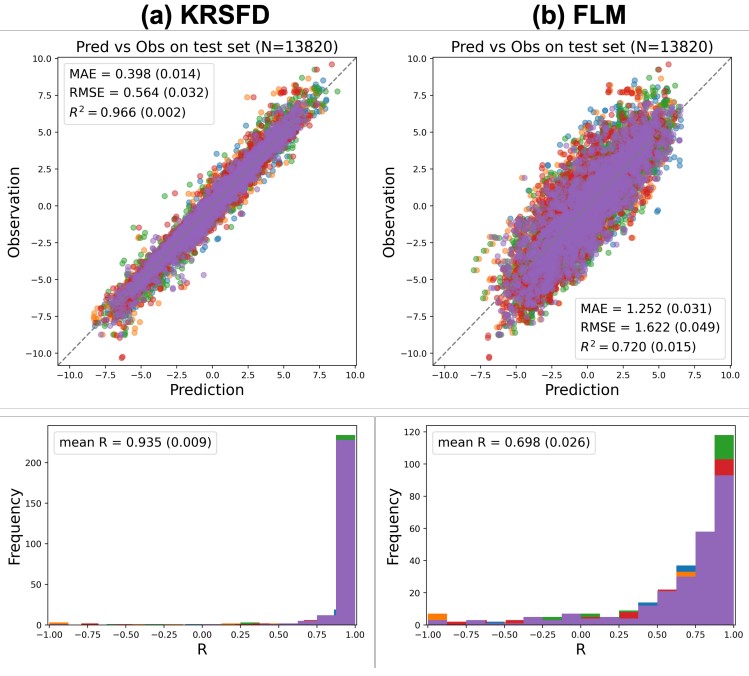

Figure 4: Prediction results for the test samples of the sparse artificial data using the (a) KRSFD and (b) FLM models. The scatter plots and histograms are color-coded in five different colors corresponding to the five independent data splitting.

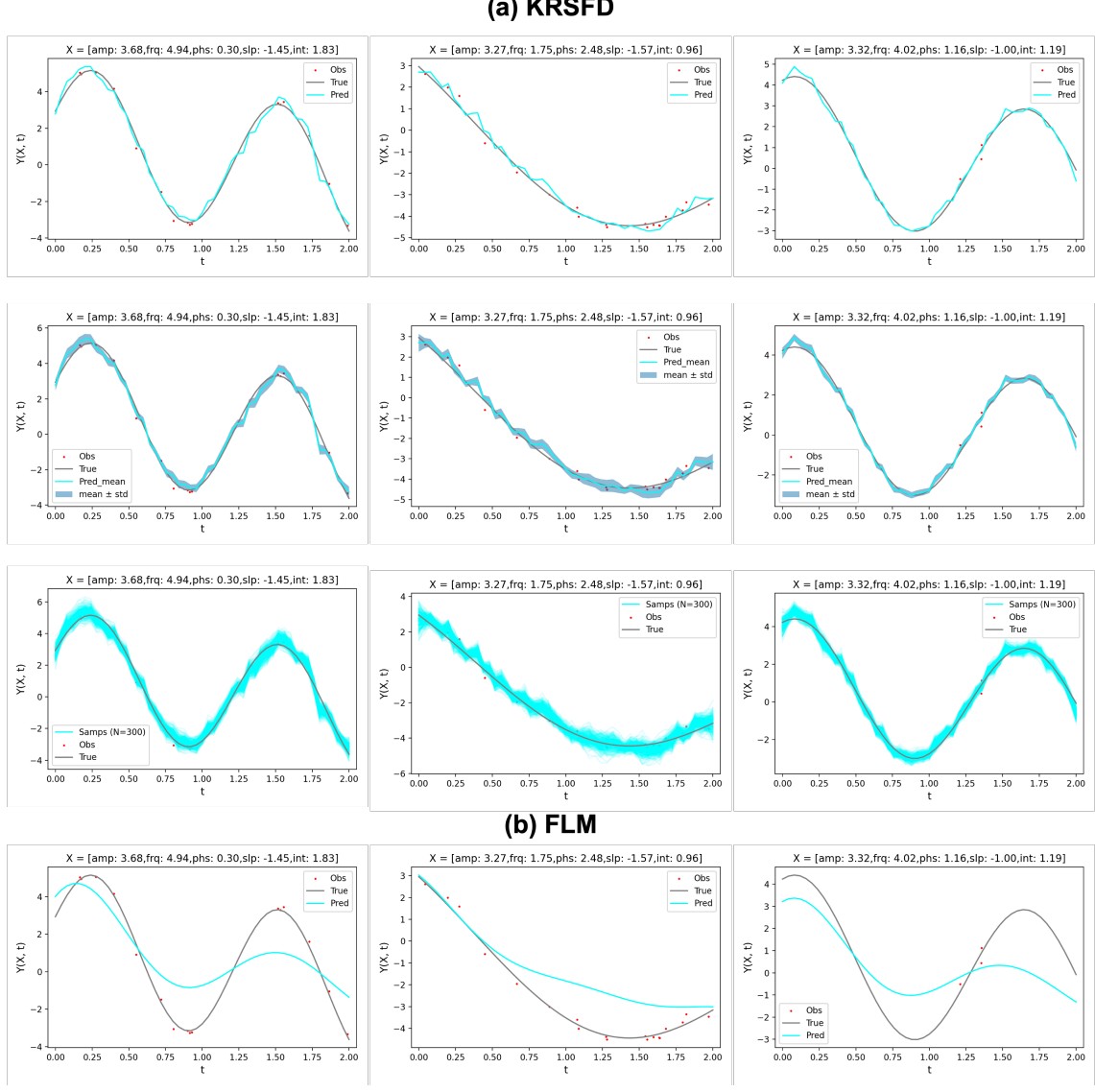

Figure 5: Functional prediction results for the test samples. The gray line shows the true data before adding observation noise, the red dots show the observed data points, and the light blue line shows the functional predictions. The title of each figure indicates the actual amplitude, frequency, phase, slope, and intercept values for that input. The first through third rows show the predictions using the KRSFD model. The first row shows only the mean of the predictive distribution, the second row shows the predicted mean ±1 standard deviation (blue band) of the predictive distribution, and the third row shows the results of sampling the function 300 times from the predictive distribution. The fourth row shows the predictions using the FLM model.

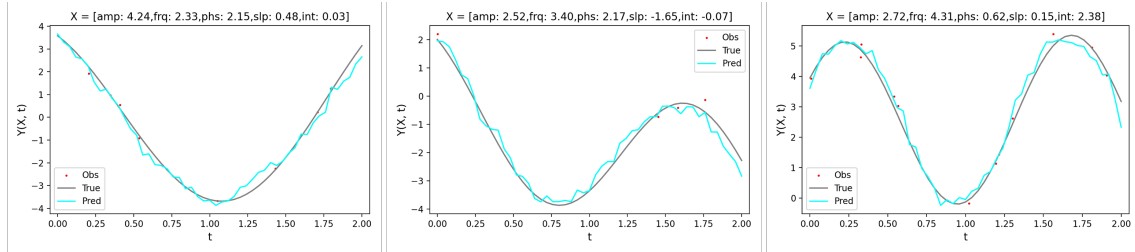

Figure 6: Functional prediction results for the training samples using the KRSFD model. The gray line shows the true data before adding observation noise, the red dots show the observed data points, and the light blue line shows the functional predictions. The title of each figure indicates the actual amplitude, frequency, phase, slope, and intercept values for that input.

## 3.3 Predicting density of states for stable metal compounds

We predicted the density of states (DOS) for stable metal compounds—key indicators of electronic properties—based exclusively on their chemical compositions. Our analysis leveraged the DOS data from the Materials Project (version released on April 21, 2024) Jain et al. (2013); Project, which was computationally derived using the density functional theory (DFT) method Hohenberg & Kohn (1964). A dataset of 13,662 DOS profiles was normalized and smoothed. We transformed the chemical composition of each compound into a 580-dimensional kernel mean descriptor (KMD) Kusaba et al. (2023) (the data processing techniques are described in Appendix D). The resulting DOS data was summarized in a $13662 \times 470$ observation matrix $Y$, a $13662 \times 580$ covariate matrix $X$, and a 470-dimensional measurement point vector $t$. The training and test inputs were randomly split in a 4:1 ratio. To calculate the standard deviation of the predictions, we prepared five randomly split training-test datasets. KRFD, FLM, and KRRs models were applied for the dataset. To efficiently manage the computational demands, various optimization strategies were implemented during the model training and hyperparameter tuning phases (see Appendix D for details).

The hyperparameter sets and solution spaces for the KRFD, FLM, and KRR models were identical to those described in Section 3.1. The optimal hyperparameter values were $\lambda_G = 6.808 \times 10^{-3}, \lambda_T = 0.011, \lambda_M = 6.798 \times 10^{-3}, \sigma_G = 28.970, \sigma_T = 4.744, \sigma_M = 29.412, k_X = Laplacian$ and $k_T = Laplacian$ for KRFD and $\lambda = 0.895, \sigma = 1.425$ and $k_T = Gaussian$ for FLM. The optimal hyperparameter values for the 470 KRR models are summarized in Figure B2.

Table 3 reports the mean and standard deviation of the MAE, RMSE, $R^2$, and mean R metrics for the five independent experiments with different data splitting. The parity plots of the predicted and observed values are shown in the top row of Figure 7. The bottom row of Figure 7 shows the histograms of the R values for each input. Examples of individual function predictions for the test samples by each model are shown in Figure 8. For the reasons discussed in Section 3.1, the functional prediction results with a $\pm 1$ band of the standard deviation and the results of functional sampling from the prediction distribution are shown in Figure 8 for KRFD. As shown in Table 3, the performance of FLM was much worse than that of the KRFD and KRR models, indicating that the relationship between the DOS of a stable metal compound and its chemical composition is highly nonlinear. KRFD slightly outperformed KRRs in all performance metrics. The hyperparameter optimization results suggest that DOS is sensitive to small changes in both energy and chemical composition, given that both $k_X$ and $k_T$ of the KRFD model and 454 out of 470 $k_X$s of the KRR models (see Figure B2) were selected to be $Laplacian$ rather than $Gaussian$. The results in Figure 8 and Table 3 show that, by selecting the appropriate model, the DOS of a stable metal structure can be predicted with a quantifiable degree of accuracy based on the structure's chemical composition information alone.

Table 3: Model performance for DOS prediction of stable metal compounds.

| Models | MAE | RMSE | $R^2$ | mean R |
|---|---|---|---|---|
| KRFD | $0.546 \times 10^{-3}$ ($\pm 0.009 \times 10^{-3}$) | $0.945 \times 10^{-3}$ ($\pm 0.019 \times 10^{-3}$) | $0.726$ ($\pm 0.008$) | $0.857$ ($\pm 0.005$) |
| KRRs | $0.557 \times 10^{-3}$ ($\pm 0.009 \times 10^{-3}$) | $0.956 \times 10^{-3}$ ($\pm 0.019 \times 10^{-3}$) | $0.721$ ($\pm 0.008$) | $0.853$ ($\pm 0.005$) |
| FLM | $0.920 \times 10^{-3}$ ($\pm 0.008 \times 10^{-3}$) | $1.449 \times 10^{-3}$ ($\pm 0.016 \times 10^{-3}$) | $0.355$ ($\pm 0.005$) | $0.617$ ($\pm 0.004$) |

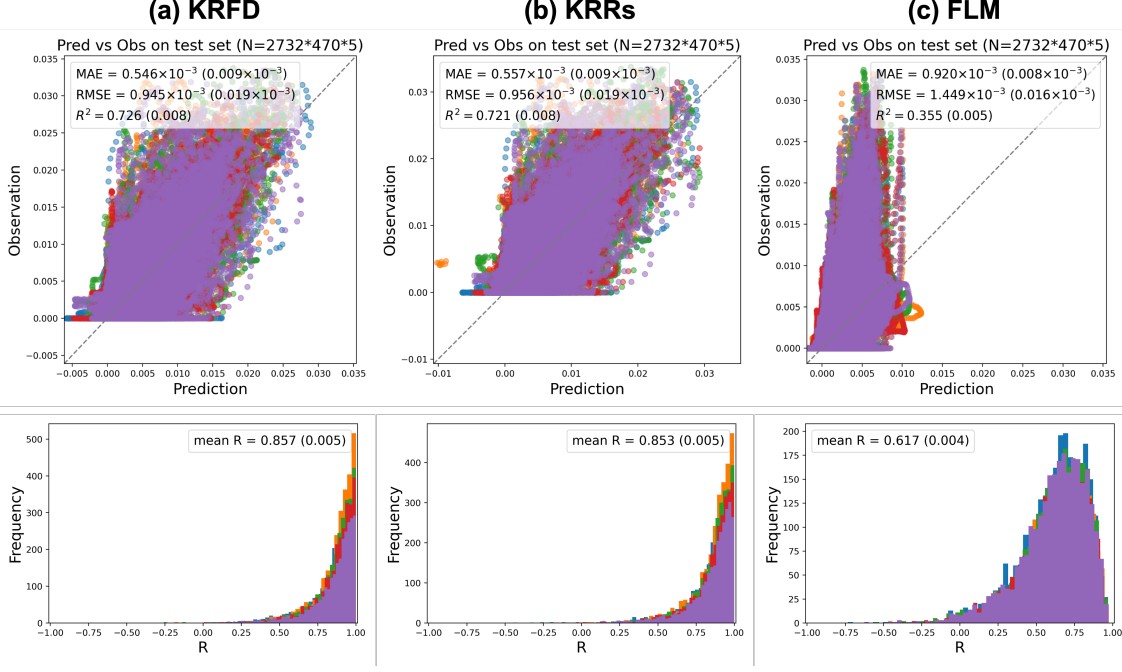

Figure 7: Prediction results for the test samples of the stable metal compound DOS data using the (a) KRFD, (b) KRRs, and (c) FLM models. The scatter plots and histograms are color-coded in five different colors corresponding to the five independent data splitting.

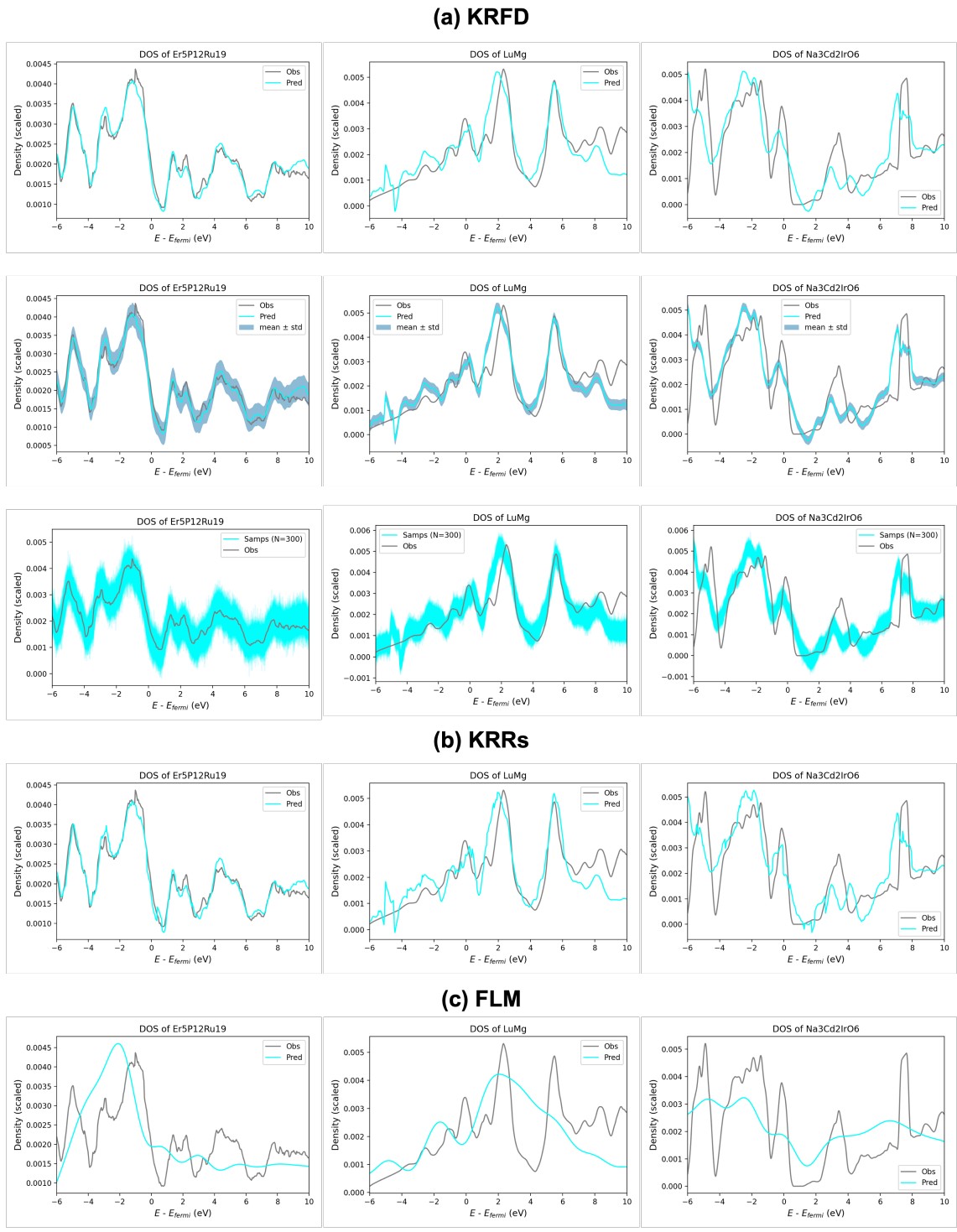

Figure 8: DOS predictions for the test samples. The gray line shows the observed DOS data, and the light blue line shows the functional prediction. The title of each figure indicates the chemical composition of that DOS profile. The first through third rows show predictions using the KRFD model. The first row shows only the mean of the predictive distribution, the second row shows the predicted mean $\pm 1$ standard deviation (blue band) of the predictive distribution, and the third row shows the results of sampling the function 300 times from the predictive distribution. The fourth and fifth rows show the predictions using the KRR and FLM models, respectively.

## 4    Conclusions

Motivated primarily by its potential application to materials data, this paper introduces KRFD, a functional output regression model based on kernel methods. Compared with FLM, the most common FSR, KRFD offers greater expressiveness by incorporating nonlinearity through kernel functions. Unlike existing nonlinear FSR, KRFD naturally handles high-dimensional nonlinearity without complicating the model structure by applying a kernel method to the covariates. This simplified form of the model enables the derivation of analytically optimal solutions, facilitates Bayesianization, and allows us to derive the mathematical expression of the model within the RKHS framework, while maintaining its expressive power. Bayesianization enables analytical quantification of prediction uncertainty and sampling of the predicted functions for any given input $X$. Viewing the KRFD through an RKHS lens shows its close connection to the MTL models based on separable kernels Bonilla et al. (2007); Ciliberto et al. (2015), suggesting a potentially broader relationship between MTL and FSR in general.

To evaluate the performance of the proposed model, numerical experiments were performed on two artificial datasets and one real-world dataset from materials science. We employed FLM and KRR as comparative models. In the three examples, KRFD outperformed the other methods in all performance metrics. FLM significantly underperformed in comparison to KRFD and KRR, demonstrating the importance of incorporating nonlinearity into models. KRR performed inferior to KRFD for the dense artificial data and DOS prediction tasks, suggesting that incorporating covariance information within $t$-space improves prediction accuracy. Unlike KRR, KRFD can be applied to sparse functional data—specifically, the model used in this case was named KRSFD. KRFD was the only model that could calculate the predictive distribution of a function, further highlighting the advantages of KRFD beyond its high prediction accuracy. Furthermore, the numerical results on the sparse functional data suggested that KRSFD can be used as a powerful nonlinear interpolation method for sparse functional data, as shown in Figure 6.

Because KRFD applies the kernel method to the covariates $X \in \mathbb{R}^p$, the memory requirements of KRFD do not depend on the dimension of covariates $p$, unlike FLM (see Appendix A). As discussed in Section 2.3, the memory requirements for KRFD are $\max\left(\mathcal{O}(N^2), \mathcal{O}(L^2)\right)$ and typically depend on $N$, because in most instances $N > L$ or $L$ can be adjusted to ensure this is the case. Thus, KRFD, like many other kernel-based machine learning models, has significant scalability challenges for large $N$. Scaling KRFD for large $N$ is a research topic for future work. To achieve this, it is expected that commonly used methods for scaling KRR such as the inducing point method Titsias (2009), Nyström approximation Williams & Seeger (2000), random Fourier features Rahimi & Recht (2007), low-rank matrix approximations Fine & Scheinberg (2001), stochastic gradient descent Bordes et al. (2009), divide-and-conquer approaches Zhang et al. (2013), and kernel approximation via spectral methods Drineas et al. (2006) could be applied. KRSFD has more serious scalability problems than KRFD. To address this issue, we have employed sparsification of the Gram matrix, the CG method, and incomplete LU decomposition, as described in Section 2.3; however, these techniques are not sufficiently effective for large data. To effectively scale KRSFD, the implementation of more advanced techniques is necessary, including the commonly used methods for scaling KRR discussed earlier (e.g., the inducing point method, Nyström approximation, and random Fourier features).

Another potential direction for future work is to enhance the flexibility of KRFD by enabling the model's kernel functions to learn during the training process. This can possibly be achieved using methods such as multiple kernel learning Gönen & Alpaydın (2011), where weights of various kernel functions are learned, or deep kernel learning Wilson et al. (2016), where kernel functions are implicitly learned within a deep neural network. Furthermore, in the Bayesian KRFD formulation, the assumption that measurement noise $\epsilon_{ij}$ follows a normal distribution independently and identically across all measurements is overly rigid. Considering more flexible noise models, such as allowing $\epsilon_{ij}$ to follow a normal distribution with varying variances $N(0, \sigma_i^2)$ for each input $X_i$, could be a direction for future research. We believe this research will advance the development of functional regression models and promote their broader application to real-world problems, including those in materials science.

## Code and data availability

The Python codes for the KRFD and KRSFD models will be made available upon publication at an anonymized repository. The codes can be used generically for functional-output regression problems. The repository will also contain all data and other codes used in this paper, allowing users to reproduce the results presented in this paper.

## Competing interests

The authors declare no conflict of interest.

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

# Appendices

## A  Details of FLM

In this study, the basis function of FLM, $\beta_j(t)$ $(j = 0, \ldots, P)$ in Eq.41, is modeled as $\beta_j(t) = \sum_{i=1}^{L} k_T(t_i, t)\theta_i^j$, where $\{t_1, \ldots, t_L\}$ are the kernel centers of $k_T(t, \cdot)$ arbitrarily defined in the $\mathbb{R}^q$ space. Consider here the problem setup described in Section 2.3. The FLM model for the observed data $Y(X_i, t_{ij})$ can then be expressed as

$$Y(X_i, t_{ij}) = \sum_{l=1}^{L} \theta_l^0 k_T(t_{ij}, t_l) + \sum_{n=1}^{p} \sum_{l=1}^{L} x_n^i \theta_l^n k_T(t_{ij}, t_l), \quad (i = 1, \ldots, N, \; j = 1, \ldots, N_i), \tag{A1}$$

where $X_i = (x_1^i, \ldots, x_p^i)^\top$.

Let $X'_i$ be the augmented vector of $X_i$ given as $X'_i = (1, x_1^i, \ldots, x_p^i)^\top$. Using the kernel $k_T(t_i, t)$, the $N_i \times L$ matrices $\boldsymbol{T_i}$ $(i = 1, \ldots, N)$ are defined in the same way as in Eq.26. Let $\boldsymbol{w}_i$ $(i = 1, \ldots, N)$ be the $N_i \times (p+1)L$ matrices given as $\boldsymbol{w}_i = X'^\top_i \otimes \boldsymbol{T_i}$ and denote $\sum_{i=1}^{N} N_i = S$. Using these matrices, we define the $S \times (p+1)L$ matrix $\boldsymbol{W}$ as

$$\boldsymbol{W} = \begin{pmatrix} \boldsymbol{w}_1 \\ \vdots \\ \boldsymbol{w}_i \\ \vdots \\ \boldsymbol{w}_N \end{pmatrix}. \tag{A2}$$

Using $\boldsymbol{W}$, Eq.A1 can be rewritten as

$$\boldsymbol{y} = \boldsymbol{W} \cdot \boldsymbol{\theta} \tag{A3}$$

where $\boldsymbol{y} = (Y(X_1, t_{11}), \cdots, Y(X_1, t_{1N_1}), \cdots, Y(X_N, t_{N1}), \cdots, Y(X_N, t_{NN_N}))^\top \in \mathbb{R}^S$, and $\boldsymbol{\theta} = (\theta_1^0, \cdots, \theta_L^0, \theta_1^1, \cdots, \theta_L^1, \cdots, \theta_1^p, \cdots, \theta_L^p)^\top \in \mathbb{R}^{(p+1)L}$.

Therefore, the objective function of FLM with a $\ell_2$ regularization can be written as

$$\min_{\boldsymbol{\theta}} \|\boldsymbol{y} - \boldsymbol{W}\boldsymbol{\theta}\|_2^2 + \lambda \|\boldsymbol{\theta}\|_2^2, \tag{A4}$$

where $\lambda$ is the regularization strength to adjust the influence of $\ell_2$ regularization.

The optimal solution of Eq.A4 is written as

$$\hat{\boldsymbol{\theta}} = (\boldsymbol{W}^\top \boldsymbol{W} + \lambda \boldsymbol{I}_{(p+1)L})^{-1} \boldsymbol{W}^\top \boldsymbol{y}, \tag{A5}$$

where $\boldsymbol{I}_{(p+1)L}$ is a $(p+1)L \times (p+1)L$ identity matrix.

Because $(\boldsymbol{W}^\top \boldsymbol{W} + \lambda \boldsymbol{I}_{(p+1)L})$ is a $(p+1)L \times (p+1)L$ matrix, the memory requirement of the FLM model is $\mathcal{O}((p+1)^2 L^2)$ in general. Because the number of kernel centers $L$ can be adjusted as desired by the user, the memory requirement of the FLM model depends on the dimension of covariates $p$. Although the FLM model is free to set the kernel centers, in Sections 3.1 and 3.3, the kernel centers were set to the fixed measurement point set for a fair comparison with the KRFD model.

## B  Hyperparameters for KRRs selected by Optuna

Figures B1 and B2 show the histograms of the hyperparameters for the KRRs models selected by Optuna in Sections 3.1 and 3.3, respectively.

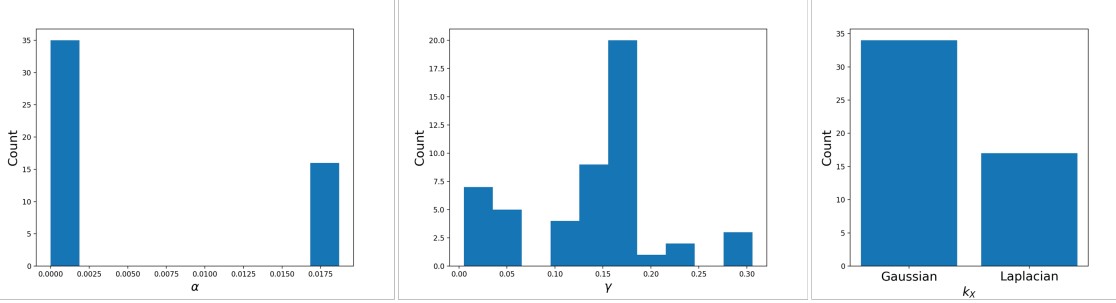

Figure B1: Histograms of the optimal hyperparameter values for the 51 single-task KRR models used in Section 3.1. The left, middle, and right histograms correspond to the regularization strength $\alpha$, inverse scale parameter $\gamma$, and type of kernel function defined in $X$-space $k_X$, respectively.

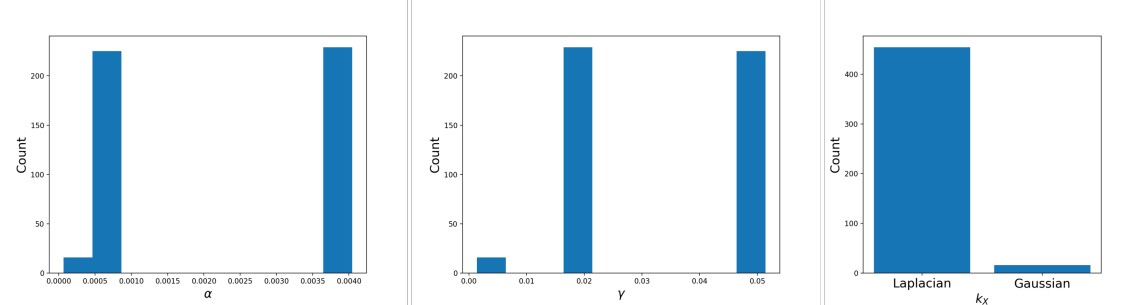

Figure B2: Histograms of the optimal hyperparameter values for the 470 single-task KRR models used in Section 3.3. The left, middle, and right histograms correspond to the regularization strength $\alpha$, inverse scale parameter $\gamma$, and type of kernel function defined in $X$-space $k_X$, respectively.

## C  Application of KRFD to the reduced dense artificial data

The total sample size of the data used in Section 3.2 was 10,962. In Section 3.1, the total sample size of the data was $1,000 \times 51 = 51,000$. To facilitate a performance comparison of KRFD and KRSFD, we performed a numerical experiment by reducing the size of the KRFD dataset to approximately match that of KRSFD.

The functional data used here were the same as that used in Section 3.1 except that the number of measurement points was reduced from 51 to 11, resulting in a total of 11,000 samples. The 11 measurement points were prepared by selecting every fifth of the 51 measurement points. Hyperparameter optimization and KRFD training was performed as described in Section 3.1. The optimal hyperparameter values were $\lambda_G = 3.433 \times 10^{-3}, \lambda_T = 1.446 \times 10^{-3}, \lambda_M = 4.738 \times 10^{-6}, \sigma_G = 1.857, \sigma_T = 1.490, \sigma_M = 6.241, k_X = Gaussian, k_T = Laplacian$. Table C1 reports the mean and standard deviation of the MAE, RMSE, $R^2$, and mean R metrics for the five independent experiments with different data splitting. The parity plot of the predicted and observed values and the histograms of the R values for each input are shown in the left and right panels of Figure C1, respectively. Examples of individual function predictions for the test samples by the KRFD model are shown in Figure C2. Comparison of the results in Table C1 with Tables 1 and 2 shows that the prediction performance of KRFD deteriorated as the data size decreased. However, it consistently outperformed KRSFD, which used a comparable amount of data, across all performance metrics.

Table C1: Model performance for the reduced dense artificial data using KRFD.

| MAE | RMSE | $R^2$ | mean R |
|---|---|---|---|
| 0.267 ($\pm$0.008) | 0.379 ($\pm$0.017) | 0.985 ($\pm$0.001) | 0.975 ($\pm$0.003) |

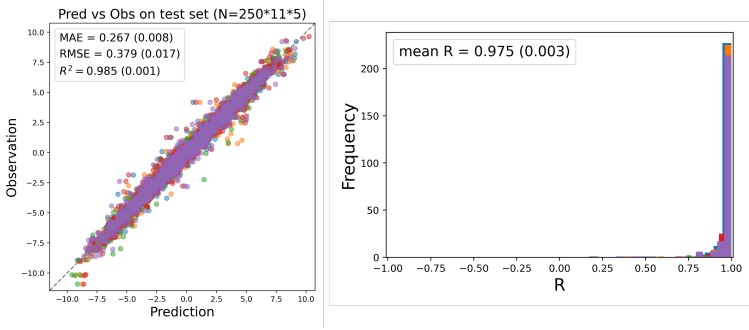

Figure C1: Prediction results for the test samples of the reduced dense artificial data using the KRFD model. The scatter plots and histograms are color-coded in five different colors corresponding to the five independent data splitting.

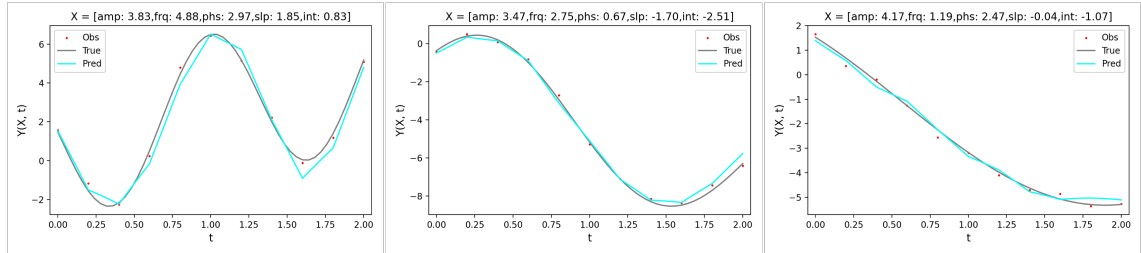

Figure C2: Functional predictions for the test samples using KRFD. The gray line shows the true data before adding observation noise, the red dots show the observed data points, and the light blue line shows the functional predictions. The title of each figure indicates the actual amplitude, frequency, phase, slope, and intercept values for that input.

# D Detailed methodologies for DOS data

## D.1 Data processing details for DOS data

In Section 3.3, we attempted to predict the DOS of stable metal compounds using only their chemical composition information. DOS describes the number of states available at each energy level that electrons can occupy in a material Kittel & McEuen (2018). It is an essential tool for understanding the electronic structure of a material, provides key insight into its physicochemical properties, and serves as an important source of information in the materials discovery process. We obtained the DFT-calculated DOS data for stable metal compounds from Materials Project (version released on April 21, 2024) Jain et al. (2013); Project.

The DOS data for stable metal compounds were available for 13,689 compounds, which are discrete data consisting of energy and density of states at that energy. We normalized the energies for each compound so that the Fermi energy for each compound took the value of zero. For each compound, these discrete data were linearly interpolated and constantly extrapolated over a range of energies from -6.5 to 10.5 eV. The interpolated data were then transformed into dense discrete functional data on 500 evenly distributed grid points ranging from -6.5 to 10.5 eV. Compounds with a maximum density greater than 500 were considered outliers and were removed from the dataset. As a result, 13,662 DOS data remained.

Because the mesh size used in the DFT calculations to obtain the DOS data in the Materials Project was too small, the DOS data were very noisy. Hence, we smoothed the DOS data for each compound by a moving average method with a window of 0.6 eV (= 17 data points). Next, to avoid bias at both edges of the data obtained by the moving average method, the data outside of -6.0 to 10.0 eV were removed. After smoothing, 470 data points remained for each compound. Finally, the DOS data were scaled so that the sum of the density values of the 470 data points for each compound was 1.

The chemical composition of each stable inorganic compound was converted into a 580-dimensional vector using the KMD methodology Kusaba et al. (2023). KMD is a method for converting a mixture system consisting of the features of each constituent element and their mixing ratios into a fixed-length vector using kernel mean embedding Muandet et al. (2016). In this example, the mixing ratio refers to the chemical composition and the features of each constituent element refer to the 58 element features (see Table I in Kusaba et al. (2023) for details).

### D.2 Optimization Techniques for Computational Efficiency

The KRFD, FLM, and KRR models were applied for the DOS data. The training and test inputs were randomly split in a 4:1 ratio. To calculate the standard deviation of the predictions, we prepared five randomly split training-test datasets. Because both $N$ and $L$ were large in this data, the following measures were taken to reduce the computational costs for model training and hyperparameter optimization.

For KRFD, the hyperparameters of the models were optimized using Bayesian optimization with Optuna. This process involved five-fold cross-validation on a randomly selected 50% ($N = 5,465$) subset of the training set from the first training-test dataset among the five randomly split datasets. The number of trials for optimization was set to 30.

Hyperparameter optimization was performed independently for each KRR corresponding to $t_i$ ($i = 1, \ldots, 470$) using Optuna. This process involved five-fold cross-validation of a randomly selected 20% ($N = 2,186$) subset of the training set from the first training-test dataset. The number of trials was set to 10 for each KRR model.

For FLM, which requires finding the inverse of the $(p + 1)L \times (p + 1)L$ matrix as in Eq. A5, we reduced the dimension of covariates $p$ from 470 to 29 using principal component analysis (PCA) Hotelling (1933) on the design matrix $X$. The top 29 components were selected so that the cumulative explained variance ratio exceeded 90% (the cumulative explained variance ratio was 90.55% at the 29th component). Hyperparameter optimization was performed using Optuna with the training set ($N = 10,930$) from the first training-test dataset. The number of trials was set to 20.

By setting the hyperparameters to their optimal values, the KRFD, KRR, and FLM models were trained using the entire training portion (100%) of each of the five randomly split datasets.

