# OpenReview forum: "Bayesian Kernel Regression for Functional Data"
_TMLR — Rejected by TMLR_

### Review · Reviewer_dYjx · 2025-04-28

**Summary Of Contributions:**

The paper presents a version of kernel regression for multivariate outputs indexed by a variable $t \in \mathbf{R}^q$. When all observations are at the same $t$ values, the resulting covariance between inputs and outputs has a Kronecker structure that can be exploited. If not, then the authors sparsify the resulting covariance matrix with truncated kernels. A maximum a posteriori estimator is used on the hyperparameters. The functional representation in RKHS is given. The authors test their regression model on two toy examples plus a test case from material science against simpler versions of the model.

**Audience:**

No

**Claims And Evidence:**

No

**Requested Changes:**

The authors should start by reading the existing literature, starting from:
- Alvarez, M. A., Rosasco, L., & Lawrence, N. D. (2012). Kernels for vector-valued functions: A review. Foundations and Trends® in Machine Learning, 4(3), 195-266.
- Wilson, A., & Nickisch, H. (2015, June). Kernel interpolation for scalable structured Gaussian processes (KISS-GP). In International conference on machine learning (pp. 1775-1784). PMLR.
- Tighineanu, P., Skubch, K., Baireuther, P., Reiss, A., Berkenkamp, F., & Vinogradska, J. (2022, May). Transfer learning with gaussian processes for bayesian optimization. In International conference on artificial intelligence and statistics (pp. 6152-6181). PMLR.
- Maddox, W. J., Balandat, M., Wilson, A. G., & Bakshy, E. (2021). Bayesian optimization with high-dimensional outputs. Advances in neural information processing systems, 34, 19274-19287.
- Vien, N. A., Zimmermann, H., & Toussaint, M. (2018, April). Bayesian functional optimization. In Proceedings of the AAAI Conference on Artificial Intelligence (Vol. 32, No. 1).
- Casenave, F., Staber, B., & Roynard, X. (2023). Mmgp: a mesh morphing gaussian process-based machine learning method for regression of physical problems under nonparametrized geometrical variability. Advances in Neural Information Processing Systems, 36, 43972-43999.

For the use of compact support kernels and sparsity:
- Terenin, A., Burt, D. R., Artemev, A., Flaxman, S., van der Wilk, M., Rasmussen, C. E., & Ge, H. (2024). Numerically stable sparse gaussian processes via minimum separation using cover trees. Journal of Machine Learning Research, 25(26), 1-36.
- Barber, J. (2020). Sparse gaussian processes via parametric families of compactly-supported kernels. arXiv preprint arXiv:2006.03673.

It should also be noted that the default method to go back to the scalar case is not to independently train regression models but to perform PCA first, as proposed for instance by Dave Higdon, Jim Gattiker, Brian Williams, and Maria Rightley. Computer model calibration using high
dimensional output. Journal of the American Statistical Association, 103(482):570–583, 2008.

When mentioning uncertainty quantification, a suitable metric should be added to the comparison.

 Typos:
P2 depedent
P4 gram
P5 maximum posteriori
P9 incorpolating  / Concequently

**Strengths And Weaknesses:**

The authors are unaware of the existing literature on kernel methods for vectorial outputs. Similar and more complex derivations are well known, in particular in the Gaussian process literature.

Despite the title teasing functional data, the work is limited to vectorial outputs, i.e., the problem is discretized.

The results on the simple synthetic test case should be compared to alternative methods, plus they are not particularly convincing on the second and third examples.

---

### Review · Reviewer_oBoy · 2025-05-01

**Summary Of Contributions:**

This paper describes a regression framework, called Kernel Regression for Functional Data (KRFD), to make enhanced predictions by leveraging correlations across functional data through a kernel-based structure. The proposed Bayesian scheme can be performed in closed form thanks to tailored conjugate priors, leading to posterior inference on the model parameters with associated uncertainty quantification and predictive distributions for the underlying functions. The proposed approach is illustrated, evaluated, and compared with two competitors, namely Kernel Ridge Regression and the Functional Linear Model, on one dense and one sparse synthetic dataset, as well as a real dataset from materials science.

**Audience:**

No

**Broader Impact Concerns:**

No particular ethical concerns.

**Claims And Evidence:**

No

**Requested Changes:**

The paper should include a comprehensive and detailed literature review, which is abundant on this topic but largely overlooked in the current version. The successive approaches, ranging from traditional linear models of coregionalisation to more recent methods of multi-output and multi-task Gaussian processes, should be presented within the context of decades of research in the field of kernel-based regression for functional, longitudinal, and spatial data. Then, the proposed method should be carefully positioned within the range of existing methods and clearly demonstrate its specificity, as in the current form of the paper, it could appear as a naive version or a particular case of multi-output GPs.

**Strengths And Weaknesses:**

In terms of strengths, the paper is well-written and concise, providing sufficient mathematical details to make it reasonably easy to follow. One interest of this approach I could acknowledge would lie in the $\max(\mathcal{O}(N^3), \mathcal{O}(L^3))$ computational complexity, though it only applies in the restricted case of aligned measurements across all functions (which is a severe limitation compared with natural alternatives).

The primary and major weakness of the present article stems from its (almost) complete oversight of the existing literature on the problem being addressed.
At least two broad communities, historically focused on *geostatistics* and subsequently on *multi-output Gaussian processes*, have developed and extended a large number of efficient modelling strategies for multivariate functional data, which are kernel-based and generally treated in a Bayesian manner.
Since those two aspects are at the centre of the proposed approach, it is puzzling to read no mention of such methods (except a brief discussion on Bonilla's paper (2007)).
To be honest, most mathematical expressions in the paper resemble those from multi-output GPs so closely that I would be really surprised if it were not a particular case of an existing piece of work.
Predictive performances are far below the state-of-the-art of modern multi-output GP methods (which are not compared with in the Experiments), and even uncertainty quantification is poorly calibrated (credible intervals in Figures 3 and 4 capture only a handful of points and are clearly underestimated).

Since I have not taken the (probably long) time to find exactly which existing model the present approach could be derived from, here is a list of, in my opinion, some critical articles on this topic.
I hope that the few papers below and related others can help authors better position their work within the literature and discover alternatives that outperform their current model, which they might try to extend or adapt to their applicative context.

- An essential historical reference, introducing the *linear model of coregionalization*, one of the simplest ways to model multiple correlated functions :
Journel, A. G., & Huijbregts, C. J. (1976). Mining geostatistics.

- The problem is also well-known and extensively studied as *co-kriging* in geostatistics :
Goovaerts, P. (1997). Geostatistics for natural resources evaluation. Oxford university press.

- A first proposal to leverage process convolution to model jointly spatial phenomena:
David M. Higdon. Space and space-time modelling using process convolutions. In C. Anderson, V. Barnett, P. Chatwin, and A. El-Shaarawi, editors,
Quantitative methods for current environmental issues, pages 37–56. Springer-Verlag, 2002

- A comprehensive and unifying review on multi-output kernels. I would probably advise authors to start from this paper for a good overview of more modern approaches on the topic :
Mauricio A. Alvarez, Lorenzo Rosasco, and Neil D. Lawrence (2012). Kernels for vector-valued functions: a review. Foundations and Trends in Machine Learning, 4(3):195–266

- Another approach is based on underlying latent variables whose correlations are captured by a kernel. This allows dimensionality reduction and leveraging the kernel trick for the output  matrix as well, instead of learning its full structure :
Dai, Z., Álvarez, M., & Lawrence, N. (2017). Efficient modeling of latent information in supervised learning using gaussian processes. Advances in Neural Information Processing Systems, 30.

- A version of multi-task GPs that differs from previous multi-output views by sharing information across multiple functions through a mean process (roughly similar to the $\mu_0$ term in your model) instead of the full covariance structure, leading to linear scaling in the number of tasks and enabling their simultaneous clustering.
Leroy, A., Latouche, P., Guedj, B., & Gey, S. (2023). Cluster-specific predictions with multi-task Gaussian processes. Journal of Machine Learning Research, 24(5), 1-49.

---

### Review · Reviewer_7nRL · 2025-05-06

**Summary Of Contributions:**

In this paper, the authors propose a kernel-based method for function-on-scalar regression.

The function-on-scalar task is characterized by learning from a series of values $\\{\\boldsymbol{x}\_i\\}{i=1}^N$ alongside another set of inputs $\\{\\boldsymbol{t}\_\\ell\\}\_{\\ell=1}^{L}$, where the goal is to predict the function $y\_{\\boldsymbol{x}}(\\boldsymbol{t})$. Common approaches in functional data analysis model this using basis expansions: $\\tilde{y}\_{\\boldsymbol{x}}(\\boldsymbol{t}) = \\sum\_{j=1}^B c\_j(\\boldsymbol{x})\\phi\_j(\\boldsymbol{t})$, where $B$ basis functions are selected or learned, and their coefficients depend on the covariates $\\boldsymbol{x}$.

The authors propose using kernel functions for both components: $\\phi\_j(\\boldsymbol{t}) = k\_T(\\boldsymbol{t}\_j,\\boldsymbol{t})$ and $c\_j(\\boldsymbol{x}) = \\sum\_{n=1}^N \\alpha\_{nj}k\_G(\\boldsymbol{x}\_n, \\boldsymbol{x})$. As a result, $y\_{\\boldsymbol{x}}(\\boldsymbol{t})$ is modeled as an element of the reproducing kernel Hilbert space (RKHS) associated with the tensor product kernel $k\_y(\\boldsymbol{x},\\boldsymbol{x}',\\boldsymbol{t},\\boldsymbol{t}') = k\_G(\\boldsymbol{x},\\boldsymbol{x}') k\_T(\\boldsymbol{t},\\boldsymbol{t}')$. The authors show how to perform MAP estimation of hyperparameters, derive the posterior distribution over weights, and adapt the representer theorem to this setting.

The model is evaluated on two synthetic datasets and a real-world task: predicting the density of states for metal compounds.

**Audience:**

No

**Broader Impact Concerns:**

I believe this work does not require adding a broader impact statement.

**Claims And Evidence:**

No

**Requested Changes:**

- **[Critical]** The authors should evaluate the quality of their predictive distribution, as developed in Section 2.2, using metrics such as marginal log-likelihood.
- **[Critical]** In light of my previous comments, I believe the authors should revise Section 2.5 to more clearly articulate the relationship between their proposal and tensor product factored kernels/multi-task learning.
- **[Critical]** I recommend an elaboration in both the text and experiments to demonstrate the utility of having the functional form of $y_{\boldsymbol{x}}(\boldsymbol{t})$ in different tasks, and what difference the proposed method makes in such settings. With this change, I believe more of TMLR’s audience would be interested in this work. Otherwise, the authors should compare their model with other regression approaches, such as Gaussian Processes with structured kernels and deep neural networks for a more balanced view of the perfomance of the proposal.
- In several cases, in-text citations (`\citet`,  `\textcite`) are used where parenthetical citations  (`\citep`,  `\parencite`) would be more appropriate. For example, in the second paragraph, all of the citations would be better presented parenthetically.

**Strengths And Weaknesses:**

In terms of methodology, the authors claim that their method is a generalization of multi-task kernel methods, such as multi-task Gaussian process (GP) models. However, their assertion that multi-task learning (MTL) models lack the capacity to handle continuous task indices is inaccurate, as models like *Kernel Multi-task Learning using Task-specific Features* (Bonilla et al., 2007) employ an equivalent tensor product kernel factorization as proposed in the paper. Additionally, the complexity benefit attributed to KRFD has been previously noted in the GP literature, for example, in works such as *Scalable Inference for Structured Gaussian Process Models* (Saatçi, 2011) and *Fast Kernel Learning for Multidimensional Pattern Extrapolation* (Wilson & Gilboa et al., 2014), and is implemented in libraries like GPyTorch (Gardner et al., 2018).

In general, I believe the experimental validation is not in line with the rest of the paper:

- The benefits of function-on-scalar regression are not clearly demonstrated in the paper or in the experimental setup. The basis functions do not appear to be evaluated based on their functional properties (e.g., integrals, derivatives), but only at specific evaluation points. In this case, the task reduces to standard regression.
- The potential advantages of Bayesian learning are not explored, as the experiments only report point-estimate metrics such as MAE and RMSE. In this sense, Section 2.2 does not appear to be empirically validated.

Additionally, both baselines—FLM and the work by Iwayama et al. (2022)—differ slightly in form from the model proposed by the authors. While the former include a $\mu(t)$ term and no $\mu(X)$ term, the authors’ model does the opposite. Any clarification on this would be worthwhile and would help better contextualize the modeling assumptions behind the compared approaches.

---

### Decision · Action_Editor_rHNj · 2025-05-15

**Recommendation:** Reject

**Comment:**

A major revision for this paper is required. Once it is done, the paper will need reviewing from scratch.

**Audience:**

There is an audience for this paper at TMLR.

**Claims And Evidence:**

The paper is unaware of key models in the literature that have already addressed this type of methodological problem. Without a consistent acknowledgement of such methods and a proper experimental comparison, there is no basis to accept the paper at this stage. The reviews provided here amount to a major revision, and I encourage the authors to reflect on whether they will want to submit a new version at a later point.

**Resubmission Of Major Revision:**

The authors may consider submitting a major revision at a later time.